# Inferring effective couplings with Restricted Boltzmann Machines

A. Decelle [1,2], C. Furtlehner [2], A. d. J. Navas Gómez [1*] B. Seoane [2]

**1** Departamento de Física Teórica, Universidad Complutense de Madrid, 28040 Madrid, Spain.
**2** Université Paris-Saclay, CNRS, INRIA Tau team, LISN, 91190, Gif-sur-Yvette, France.
* alfonn01@ucm.es

January 25, 2024

## Abstract

Generative models offer a direct way of modeling complex data. Energy-based models attempt to encode the statistical correlations observed in the data at the level of the Boltzmann weight associated with an energy function in the form of a neural network. We address here the challenge of understanding the physical interpretation of such models. In this study, we propose a simple solution by implementing a direct mapping between the Restricted Boltzmann Machine and an effective Ising spin Hamiltonian. This mapping includes interactions of all possible orders, going beyond the conventional pairwise interactions typically considered in the inverse Ising (or Boltzmann Machine) approach, and allowing the description of complex datasets. Earlier works attempted to achieve this goal, but the proposed mappings were inaccurate for inference applications, did not properly treat the complexity of the problem, or did not provide precise prescriptions for practical application. To validate our method, we performed several controlled inverse numerical experiments in which we trained the RBMs using equilibrium samples of predefined models with local external fields, 2-body and 3-body interactions in different sparse topologies. The results demonstrate the effectiveness of our proposed approach in learning the correct interaction network and pave the way for its application in modeling interesting binary variable datasets. We also evaluate the quality of the inferred model based on different training methods.

# 1   Introduction

In recent years, generative models have become increasingly popular in machine learning (ML). These models can learn complex statistical patterns from data sets and generate new data that closely resembles the original. Among the various generative tools, energy-based models (EBMs) [1–5] stand out for their unique capability of systematic modeling. EBM training consists of finding the set of model parameters $\boldsymbol{\theta}$ that best encodes the empirical distribution of $\boldsymbol{x}$ given by the data set into the Boltzmann-Gibbs distribution associated with an energy function $\mathcal{H}_{\boldsymbol{\theta}}(\boldsymbol{x})$, i.e,

$$p_{\boldsymbol{\theta}}(\boldsymbol{x}) = \frac{1}{\mathcal{Z}_{\boldsymbol{\theta}}} e^{-\mathcal{H}_{\boldsymbol{\theta}}(\boldsymbol{x})}, \quad \text{with} \quad \mathcal{Z} = \sum_{\boldsymbol{x}} e^{-\mathcal{H}_{\boldsymbol{\theta}}(\boldsymbol{x})},$$

where $\sum_{\boldsymbol{x}}$ runs over all possible combinations of the variables so that the distribution is properly normalized, and $\mathcal{Z}$ is the partition function. This means that once the EBM is trained, its energy function $\mathcal{H}_{\boldsymbol{\theta}}$ serves as an *effective model* for the data under study. As a physicist, one is tempted to analyze the learned neural network as a physical model that encodes complex interactions between the data variables and use it to gain valuable insights into its internal rules or building blocks, i.e. to *learn* from the data. In other words,

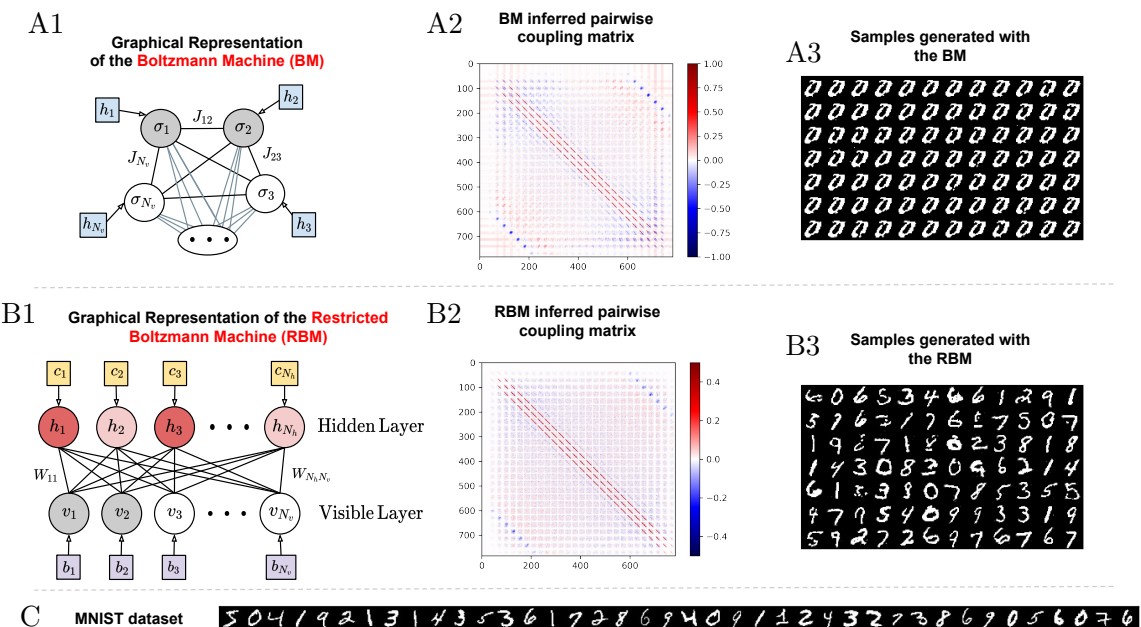

Figure 1: **Limitations of the Boltzmann Machine.** In this figure, we compare the inferred pairwise interaction coupling matrix and the samples generated by the Boltzmann Machine (BM) and the restricted Boltzmann Machine (RBM) when trained with maximum likelihood on the MNIST dataset [12]. The generated samples are obtained by iterating 84 independent Markov Chain Monte Carlo (MCMC) simulations from random initialization until convergence. In **A1** we illustrate the architecture of the BM and in **A2** the coupling matrix $J$ learned at the end of training. In **A3** we show 84 equilibrium samples of the same model. In **B1** we show the RBM architecture. In **B2** the effective pairwise interactions obtained by mapping the RBM into a generalized Ising model (using Eq. (13)) and 84 RBM equilibrium samples in **B3**. In **C** we show 40 examples from MNIST.

EBMs offer a unique opportunity to systematically extract understandable information from large amounts of data. However, this requires the use of neural network models that are simultaneously meaningful enough to capture the full complexity of the data, but also simple enough to allow a certain degree of analytical treatment.

In parallel, the goal of deriving good microscopic models whose equilibrium configurations correctly reproduce the statistics of a dataset has been addressed by physicists for decades in the context of the so-called inverse Ising problem [6]. In contrast to traditional statistical physics, which derives macroscopic properties from simple microscopic rules, the inverse Ising problem aims to infer coupling matrices and external fields from the pairwise correlations or variable frequencies observed in the data samples. This approach has successfully modeled various complex systems, including the reconstruction of synaptic connections in natural neural networks [7], the prediction of protein folding structures [8], inference of fitness landscapes in genome evolution [9,10] or gene regulatory networks [11].

Despite the straightforward interpretation of the Ising model –or the Boltzmann machine (BM), as it is better known in computer science [13] (Fig.1–A1)– as a physical model of interacting spins, BMs have a significant drawback when modeling complex binary data distributions: They can only capture up to pairwise correlations between variables and are blind to higher-order correlations. For example, when trained on the MNIST dataset [14] (a dataset of scanned images of handwritten digits, see Fig. 1–C), a trained BM struggles

to generate distinct digits, as shown in Fig. 1–A3.

There is no fundamental reason why many-body interactions should not be important for the correct description of complex arbitrary data, and omitting them in the model not only harms its generative power but could also compromise the quality of inference even at the pairwise coupling level. A possible solution to overcome the BM limitation would be to formulate the probabilistic model instead in the form of a *generalized* Ising model (GIM), which also takes into account higher-order interactions between Ising spins $\sigma_j \in \{-1, 1\}$ variables, where $j = 1, \ldots N$ is the spin index, i.e.

$$\mathcal{H}_{\mathrm{GIM}}(\boldsymbol{\sigma}) = -\sum_j H_j \sigma_j - \sum_{j_1 > j_2} J^{(2)}_{j_1 j_2} \sigma_{j_1} \sigma_{j_2} - \sum_{j_1 > j_2 > j_3} J^{(3)}_{j_1 j_2 j_3} \sigma_{j_1} \sigma_{j_2} \sigma_{j_3} + \ldots \qquad (1)$$

where $J^{(n)}_{j_1 \ldots j_n}$ is the tensor encoding the $n$-body interactions between spins with indices $j_1, \ldots, j_n$, and $H_j$ is the external field acting on spin index $j$. Again, as in the inverse Ising problem, we can try to derive all these parameters from observing a set of equilibrium configurations. While this solution would allow us to describe richer datasets, it becomes very expensive to train because the number of parameters quickly diverges as the order of the interactions increases. However, it is important to stress that the vast majority of these $J^{(n)}_{j_1 \ldots j_n}$ in (1), will be zero in real-world scenarios. Typical interaction models are often short-range and/or very sparse. The difficulty is that we do not know in advance which parameters should vanish.

A suitable alternative is the so-called Restricted Boltzmann Machine [2] (RBM, Fig. 1–B1) [1]. Returning to our example of generating MNIST digits, an RBM with a similar number of parameters as the BM shown in Fig. 1–A2 can easily learn the MNIST dataset and generate samples that resemble it (see Fig. 1–B3). In this paper, we will show that RBMs are indeed as interpretable as BMs. For example, we can extract the effective pairwise coupling parameters, as shown in Fig. 1–B2, and even go beyond 2-body interactions. As for our work here, the RBM energy function can be directly mapped into a generalized Ising-like model such as the one in Eq. (1) [15, 16], where the number of parameters to be learned is much smaller than if all $J^{(n)}_{j_1, \ldots, j_n}$ had to be learned one-by-one, since only non-zero couplings need to be encoded in the model. Thanks to this equivalence, the RBM can be understood as an effective spin-interaction model for all effects and can thus be used as an inverse Ising model with high-order interactions.

An RBM is a kind of EBM defined over a bipartite graph consisting of two layers of variables (Fig. 1–B1): the role of the interacting spins (i.e., the data variables) is played by the *visible* variables $\boldsymbol{v} = \{v_j\}_j^{N_\mathrm{v}}$, while the *hidden* variables $\boldsymbol{h} = \{h_i\}_i^{N_\mathrm{h}}$ allow us to encode the interactions between the visible variables. In this sense, as we will see, the many-body interactions are here encoded in the marginal distribution of the visible variables of the model, i.e. in $p(\boldsymbol{v}) = e^{-\mathcal{H}(\boldsymbol{v})}/Z \equiv \sum_{\boldsymbol{h}} e^{-H(\boldsymbol{v}, \boldsymbol{h})}/Z$, where $H(\boldsymbol{v}, \boldsymbol{h})$ is defined in the next equation. In the following, we will use the index $i$ for the hidden variables and $j$ for the visible units. In the RBMs considered here, both the visible and hidden ones are defined via the binary support $\{0, 1\}$, which is often referred to as Bernoulli RBM. The RBM energy function is therefore defined as

$$H(\boldsymbol{v}, \boldsymbol{h}) = -\sum_{i=1}^{N_\mathrm{h}} \sum_{j=1}^{N_\mathrm{v}} h_i W_{ij} v_j - \sum_{j=1}^{N_\mathrm{v}} b_j v_j - \sum_{i=1}^{N_\mathrm{h}} c_i h_i, \qquad (2)$$

where $\boldsymbol{W}$ is the coupling matrix, and $\boldsymbol{b}$ and $\boldsymbol{c}$ are the visible and hidden bias, respectively. A detailed definition of the model and its training procedure can be found in the Appendix A.

In contrast to deep neural networks, which are often a *black box* difficult to analyze, the simple structure of the RBM allows a high degree of analytical treatment (see e.g. a review about mean-field results [17]). For example, the phase diagram was obtained in different regions [18–20], the process of pattern encoding is well understood in the initial stages of the learning process [17, 21], it is possible to derive the TAP equations for this model, which allows easy identification of the minima of the free energy [22, 23]. These minima are used, for example, to delimit families in data sets [24].

As argued above, in this paper we extend the list of the applications of the RBMs by explicitly rewriting the RBM as a GIM. We show that these expressions enable accurate inference of the coupling constants in the model, up to a potentially arbitrary order. In practice, computing couplings beyond 4th order is challenging due to the enormous number of possible couplings to compute. However, this mapping makes it possible to combine the generative capabilities of the RBM with the direct interpretability of the learned model, thus outperforming BMs by being able to perform both tasks in virtually any dataset. This improvent comes also essentially at zero computational cost, because training an RBM (with maximum likelihood) is in generally faster than training an BM as discussed in Appendix G.

We have structured the paper as follows: In Section 2 we discuss the mapping between an RBM and models of interacting particles. We first discuss previous attempts to establish this connection, and then in Section 2.2 we derive our formal mapping between an RBM and a GIM, and a practical implementation for inference applications. In Section 3 we discuss the numerical experiments we performed to test the reliability of these expressions, in Section 3.5 we compare with previous mappings, and in Section 3.6 we discuss the implications for the quality of inference with respect to the training procedure. In Section 4 we discuss the limitations of the current approach and in Section 5 the conclusions and perspectives of this work.

## 2 The RBM as a model of interacting spins

### 2.1 Previous attempts

Several previous studies have explored the exact link between models with interacting spins and RBMs [15, 16, 25, 26]. For example, a simplified RBM (where the hidden units are Gaussians and the visible units are binary) exhibits the same thermodynamics as Hopfield networks [25]. In Refs. [15, 16] it was shown that it is possible to derive an exact mapping between a Bernoulli RBM and GIMs formulated in terms of $\boldsymbol{\sigma} = \{0, 1\}$-variables (that is, generalized *lattice gas* model). We provide an alternative (and shorter) derivation of this equivalence in Appendix B.

In Ref. [16], the authors also verified that the finite-size statistics of the samples generated by an RBM with random weights are the same as those obtained with the corresponding lattice gas model. However, the reverse test, i.e., using RBMs to correctly infer the lattice gas model that had generated the training samples, was not investigated in this work. Cossu et al. [15] proposed to use this $\{0, 1\}$ mapping to derive effective couplings, not the ones of the equivalent lattice gas model but those of the equivalent $\sigma \in \{\pm 1\}$ Ising model. In Ref. [27], the same idea was used to propose an exact construction of an ideal RBM perfectly matching a given GIM model up to 3-body interactions.

The mapping proposed by Cossu et al. [15] has serious limitations on a learned RBM since the higher order interactions that are necessarily present in the equivalent $\{0, 1\}$

model learned by the RBM[1] introduce interactions between variables at all lower orders when translated into $\sigma \in \{\pm 1\}$ spins. To achieve a reliable mapping between the RBM and a GIM, all of these additional terms must be carefully considered. This derivation has been done correctly in a very recent theoretical paper in [26], but the expressions obtained are not easy to implement in practice for interference applications, and have not been tested numerically in controlled inverse experiments yet.

The development of a correct mapping between the RBM and the GIM parameters is important not only to reproduce the model with a precise definition of the spin variables, but it is crucial to propose an accurate procedure to infer effective couplings for binary datasets in general. Indeed, the expressions obtained with the generalized lattice gas model mapping (derived in Refs. [15, 16] and Appendix B) perform very poorly in controlled inverse experiments, even when working with a rather high number of training samples. This failure is due to the tiny cost of adding high-order couplings into the effective, given that only couplings between all 1-tuples of the spin have a non-zero energy cost. This leads to expressions for the effective couplings that are very unstable for small variations of the RBM parameters and often predict strong (non-existent) interactions. This problem cannot occur in the GIM since all high-order terms contribute to the energy of a configuration. Another limitation of this expression is that it seems to be valid for a very specific (and very sparse) kind of RBMs. As we will discuss in Appendix D, the mapping between RBMs is not bijective and many different RBMs can match identical GIM models. Typical trained RBMs are not at all sparse and therefore the mapping yields very inaccurate expressions in practice. We will illustrate the shortcomings of the RBM-$\{0, 1\}$ lattice gas mapping in a later controlled experiment in Section 3.5.2, and discuss the connection with the sparse RBM in Appendix C.

In this paper, we provide (i) a precise recipe to write a given RBM as a GIM in its full complexity, (ii) a computer code to do this in practice with a set of trained parameters (see GitHub repository), and (iii) an extensive set of numerical experiments that prove the reliability, stability, and accuracy of our approach to infer the data generation model.

## 2.2 From the RBM to the generalized Ising Model

In this section, we will expand the energy function of the RBM given by Eq. (2), to write it in terms of a generalized Ising-like Hamiltonian given by Eq. (1), where $n \leq N_{\rm v}$ and $\sigma_j \in \{-1, 1\}$ (unlike the RBM variables $\boldsymbol{v}$ and $\boldsymbol{h}$, which are $\{0, 1\}$). This allows us to interpret the learned features of an RBM in terms of local many-body interactions.

Consider the marginal probability distribution of the RBM on the visible nodes, where the energy function $\mathcal{H}(\boldsymbol{v})$ is given by

$$p(\boldsymbol{v}) = \frac{1}{\mathcal{Z}} \sum_{\boldsymbol{h}} e^{-H(\boldsymbol{v}, \boldsymbol{h})} = \frac{e^{-\mathcal{H}(\boldsymbol{v})}}{\mathcal{Z}}. \tag{3}$$

Replacing the definition of the RBM energy function given by Eq. (2) in the above equation and then solving for $\mathcal{H}(\boldsymbol{v})$, one obtains

$$\mathcal{H}(\boldsymbol{v}) = -\sum_j b_j v_j - \sum_i \ln \sum_{h_i} \exp\left(c_i h_i + \sum_j h_i W_{ij} v_j\right). \tag{4}$$

---

[1]RBMs are trained on datasets with a finite number of entries, which means that they can only learn approximate models for the data. This means that the effective GIM will contain interactions up to much higher orders than those present in the model used to generate the samples.

Next, we define the change of variables

$$\sigma_j \equiv 2v_j - 1, \ \tau_i \equiv 2h_i - 1 \tag{5}$$

and the parameters $\eta_j \equiv \frac{1}{2}\left(b_j + \frac{1}{2}\sum_i W_{ij}\right)$, $\zeta_i \equiv \frac{1}{2}\left(c_i + \frac{1}{2}\sum_j W_{ij}\right)$, $w_{ij} \equiv \frac{1}{4}W_{ij}$, to write the Eq. (4) as

$$\mathcal{H}(\boldsymbol{\sigma}) = -\sum_j \eta_j \sigma_j - \sum_i \ln \cosh\left(\sum_j w_{ij}\sigma_j + \zeta_i\right). \tag{6}$$

To obtain an analytic expression that is easier to handle, we introduce the auxiliary variables $\sigma'_j$ defined over the binary support $\{-1, 1\}$. Now, it is clear that given a fixed configuration of spins $\boldsymbol{\sigma}$ the following identity

$$\sum_{\boldsymbol{\sigma}'} \prod_j \delta_{\sigma_j \sigma'_j} = 1$$

holds, so we can rewrite the Eq. (6) as

$$\mathcal{H}(\boldsymbol{\sigma}) = -\sum_j \eta_j \sigma_j - \sum_{\boldsymbol{\sigma}'} \prod_j \delta_{\sigma_j \sigma'_j} \sum_i \ln \cosh\left(\sum_j w_{ij}\sigma'_j + \zeta_i\right). \tag{7}$$

Since the the Kronecker $\delta_{\sigma_j \sigma'_j}$ can be expressed in terms of $\sigma_j$ variables as $\delta_{\sigma_j \sigma'_j} = \frac{1}{2}(1 + \sigma_j \sigma'_j)$, the Hamiltonian can be rewritten as

$$\mathcal{H}(\boldsymbol{\sigma}) = -\sum_j \eta_j \sigma_j - \frac{1}{2^{N_v}} \sum_{\boldsymbol{\sigma}'} \prod_j \left(1 + \sigma_j \sigma'_j\right) \sum_i \ln \cosh\left(\sum_j w_{ij}\sigma'_j + \zeta_i\right). \tag{8}$$

Expanding the right-hand side of the above equation, and comparing it term-by-term to Eq. (1), we found that they are equivalent if the following identifications are applied:

$$J^{(n)}_{j_1 \dots j_n} = \frac{1}{2^{N_v}} \sum_{\boldsymbol{\sigma}'} \sum_i \sigma'_{j_1} \dots \sigma'_{j_n} \ln \cosh\left(\sum_j w_{ij}\sigma'_j + \zeta_i\right), \tag{9}$$

for the $n$-body interaction couplings, and

$$H_j = \eta_j + \frac{1}{2^{N_v}} \sum_{\boldsymbol{\sigma}'} \sum_i \sigma'_j \ln \cosh\left(\sum_k w_{ik}\sigma'_k + \zeta_i\right), \tag{10}$$

for the external fields. Since the expressions given in Eqs. (9) and (10) contain the sum $\sum_{\boldsymbol{\sigma}'}$ running over the $2^{N_v}$ possible configurations of $\boldsymbol{\sigma}'$, we need to put these equations into a form suitable for evaluation. To do this, let us separate the variables $\sigma'_{j_\mu}$, with $1 \leq \mu \leq n$, involved in front of the log from the others and introduce the set of random variables

$$X^{(j_1 \dots j_n)}_i \equiv \sum_{\mu=n+1}^{N_v} w_{ij_\mu}\sigma'_{j_\mu}. \tag{11}$$

Considered themselves as random variables, the $\sigma'_{j_\mu}$ are i.i.d., uniformly over the binary support $\{-1, 1\}$. Consequently, the new variables (11) have tractable distributions to

average over. Therefore, rewriting the expression of the interaction factor with the external field $H_j$, given in the Eq. (10), in terms of an average over $X_i^{(j_1 \dots j_n)}$ gives us

$$H_j = \eta_j + \frac{1}{2} \sum_i \mathbb{E}_{X_i^{(j)}} \left[ \ln \frac{\cosh\left(\zeta_i + w_{ij} + X_i^{(j)}\right)}{\cosh\left(\zeta_i - w_{ij} + X_i^{(j)}\right)} \right]. \tag{12}$$

By following a similar procedure for the expression for the 2-body couplings, we obtain

$$J_{j_1 j_2}^{(2)} = \frac{1}{4} \sum_i \mathbb{E}_{X_i^{(j_1 j_2)}} \left[ \ln \frac{\begin{array}{c}\cosh\left(\zeta_i + w_{ij_1} + w_{ij_2} + X_i^{(j_1 j_2)}\right) \\ \times \cosh\left(\zeta_i - (w_{ij_1} + w_{ij_2}) + X_i^{(j_1 j_2)}\right)\end{array}}{\begin{array}{c}\cosh\left(\zeta_i + (w_{ij_1} - w_{ij_2}) + X_i^{(j_1 j_2)}\right) \\ \times \cosh\left(\zeta_i - (w_{ij_1} - w_{ij_2}) + X_i^{(j_1 j_2)}\right)\end{array}} \right]. \tag{13}$$

More generally, the expression for the $n$-body interaction couplings involves the averaging of $2^n$ terms w.r.t. one single $X_i^{(j_1 \dots j_n)}$ variables, and it is given by

$$J_{j_1 \dots j_n}^{(n)} = \frac{1}{2^n} \sum_i \mathbb{E}_{X_i^{(j_1 \dots j_n)}} \left[ \sum_{\sigma'_{j_1} = \pm 1} \cdots \sum_{\sigma'_{j_n} = \pm 1} \sigma'_{j_1} \dots \sigma'_{j_n} \ln \cosh \left( \sum_{\mu=1}^{n} w_{ij_\mu} \sigma'_{j_\mu} + X_i^{(j_1 \dots j_n)} + \zeta_i \right) \right]. \tag{14}$$

The above expressions can be easily implemented if the central limit theorem holds, because if $N_{\mathrm{v}} \gg n$, then $X_i^{(j_1 \dots j_n)}$ can be approximated by a normal distribution with center in 0 and variance $\sum_{j \neq j_1 \dots j_n} w_{ij}^2$ and the expected values in Eqs. (12), (13) and (14) can be easily calculated as numerical integrals. A practical implementation of this method can be found in the code in Github. This Gaussian approximation can lead to problems if, for example, some $w_{ij}$ are much larger than the rest, which means that the distribution of these $w_{ij}$ has heavy tails, or if the matrix $\boldsymbol{w}$ is very sparse. In practice, when learning the sparse interacting models studied in this paper, we never obtain sparse $\boldsymbol{w}$ (see the examples and the discussion in Appendix E). Nevertheless, for very sparse matrices the expressions (9) are tractable, which means that no approximation is needed. In Appendix E we discuss the validity of the Gaussian approximation and provide an alternative expression using Large Deviation Theory. In the following, we will only use the (12)-(14) formulas discussed in the main text, as we have found that the extraction of the effective couplings using the formulas in the appendix can be quite slow and has problems with convergence. We also want to stress that the calculation of $J_{j_1 \dots j_n}^{(n)}$ becomes intractable for large $n$ at some point (e.g. $n \gtrsim 30$). However, we are talking here about very high-order interactions which hardly seem to be applicable in practice.

In Appendix D we discuss the reverse mapping: Given a GIM, how can you construct an RBM that corresponds exactly to the same energy function? As we show there, it is relatively easy to create a perfect RBM with a sparse construction in which hidden nodes only couple visible units that interact directly with each other. However, in contrast to the unique mapping from an RBM to a GIM described in this section, there are many RBMs corresponding to a given GIM, and the sparse constructions are very rare in the huge number of possible constructions that combinatorics allows. This analysis also allows us to estimate the sufficient number of hidden nodes required to exactly reproduce a given GIM, which essentially scales with the number of non-zero couplings one needs to encode.

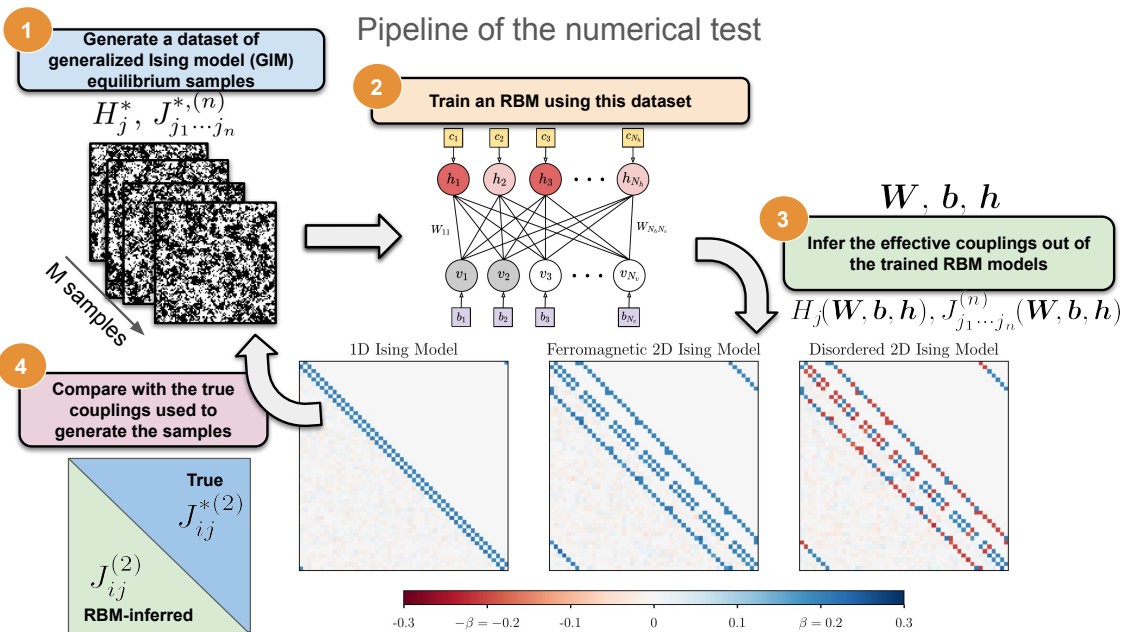

Figure 2: **Pipeline of numerical analysis.** We show a sketch of the numerical inverse Ising procedure we use to test our mapping between the RBM and a generalized Ising model (GIM) defined in Eq. (1). First, we generate equilibrium samples with a predefined GIM. Then, we train an RBM with these samples and use the RBM parameters to infer the effective fields and couplings between spins. Finally, we compare the derived couplings to the true couplings used to create the dataset. As an example, we show the comparison of the inferred and the original pairwise coupling matrices in the triangles below and above the diagonal, respectively, in three different inverse Ising experiments where the configurations in the training set were generated at $\beta = 0.2$ with the (a) 1D ferromagnetic Ising model, (b) 2D Ising model and (c) a disordered 2D Ising model (the Edwards-Anderson model) containing both positive and negative interactions.

## 3 Numerical Experiments

In the previous section, we have shown that it is possible to write all GIM parameters as a function of the RBM parameters (i.e., the $W$ matrix and the visible and hidden bias $\boldsymbol{b}$ and $\boldsymbol{c}$, respectively). In this section, we will perform systematic inverse experiments to evaluate the quality of the inferred effective model parameters with respect to the parameters used to generate the data. This approach is usually known in statistical mechanics as the inverse Ising problem [6, 28]. The pipeline of the "experimental test" used to validate our mapping can be found in Fig 2.

With this goal, we first created multiple datasets whose samples are statistically independent configurations of $N$ binary spins $\sigma = \pm 1$ distributed according to the Gibbs-Boltzmann distribution

$$p_{\mathcal{D}}(\boldsymbol{\sigma}) = \frac{1}{\mathcal{Z}_{\mathcal{D}}} e^{-\beta \mathcal{H}_{\mathcal{D}}(\boldsymbol{\sigma})}, \quad \text{with} \quad Z_{\mathcal{D}} = \sum_{\boldsymbol{\sigma}} e^{-\beta \mathcal{H}_{\mathcal{D}}(\boldsymbol{\sigma})}, \tag{15}$$

where $\beta = 1/T$ is the inverse temperature, and $\mathcal{H}_{\mathcal{D}}$ is our predefined GIM model whose parameters we will try to infer using Eqs. (12)-(14). We refer to this $\mathcal{H}_{\mathcal{D}}$ as our *ground truth* GIM. For the following numerical experiments, we will consider GIMs containing at

most third-order interactions, i.e.,

$$\mathcal{H}_{\mathcal{D}}(\boldsymbol{\sigma}) = -\sum_{\langle j_1, j_2, j_3 \rangle} J^{*(3)}_{j_1 j_2 j_3} \sigma_{j_1} \sigma_{j_2} \sigma_{j_3} - \sum_{\langle j_1, j_2 \rangle} J^{*(2)}_{j_1 j_2} \sigma_{j_1} \sigma_{j_2} - \sum_j H^*_j \sigma_j. \tag{16}$$

where the sums run only once per each interacting pair or triplet. In the following, we highlight the original model parameters with an asterisk * to distinguish them from those derived from the RBM, which are given without the asterisks. In particular, we will consider the following different ground-truth models:

1. The pure 1D Ising model at different temperatures $\beta$. We discuss the effect of the size of the training set $M$ and on the number of hidden nodes $N_{\mathrm{h}}$.

2. The 1D Ising model with 3-body interactions.

3. The 2D Ising model at different $\beta$ above and below the ferromagnetic transition.

4. Two disordered models: an Edwards-Anderson spin-glass model in 2D containing mixed $\pm 1$ interactions, and a spin glass model on a random graph with random and continuous mixed interactions.

5. We illustrate the limitations of the previous mappings when we try to infer the parameters of the 1D and 2D Ising model using Ref. [15], or those of the 1D lattice gas model using the formulas of Ref. [16].

6. We discuss the effects of the out-of-equilibrium training effects in the inference of couplings in the 2D Ising model.

Note that both the ground truth model parameters and $\beta$ fully determine the equilibrium statistics of the model, but only via their product: $\beta J^{*(2)}_{j_1 j_2}$ and $\beta H^*_j$, implying that only these joint products could be recovered by the RBM. In other words, in a perfect recovery, the RBM parameters should be $J^{(2)}_{j_1 j_2} = \beta J^{*(2)}_{j_1 j_2}$.

To quantify the quality of the inferred model, we compute the mean-squared error of the inferred parameters normalized by the mean squared value of all parameters to avoid having scores proportional to $\beta$. The error in the pairwise couplings is then

$$\Delta_{J^{(2)}} = \sqrt{\frac{\sum_{j_1 > j_2} \left( J^{(2)}_{j_1 j_2} - \beta J^{*(2)}_{j_1 j_2} \right)^2}{\sum_{j_1 > j_2} \beta J^{*(2)}_{j_1 j_2}{}^2}}. \tag{17}$$

Analogous scores can be defined for the fields $\Delta_H$ and the 3-body coupling tensor $\Delta_{J^{(3)}}$. In cases when $\beta J^{*(n)}_{j_1 j_2} = 0$ we used the root-mean-square error as error scores. To verify the correct estimation of the strength of the interaction $\beta$ in most cases, we also show non-normalized histograms of the obtained parameters. In all cases, the indices of the largest (in absolute value) inferred couplings, always coincide with those of the true non-zero couplings in the truth model. In other words, the ROC (Receiver Operating Characteristic) curve of our RBM predictions is always that of the perfect classifier. Since they are not very informative between different experiments, we will never show these curves in this paper. This also means that a high error is usually associated with a moderate noise level in the non-interacting couplings and a poor estimate of $\beta$ and not with an incorrect determination of the connectivity matrix.

Samples for the dataset are generated via Heat Bath, Metropolis-Hastings, or Wolff MCMC simulations. Details of these simulations, as well as the tests performed to ensure

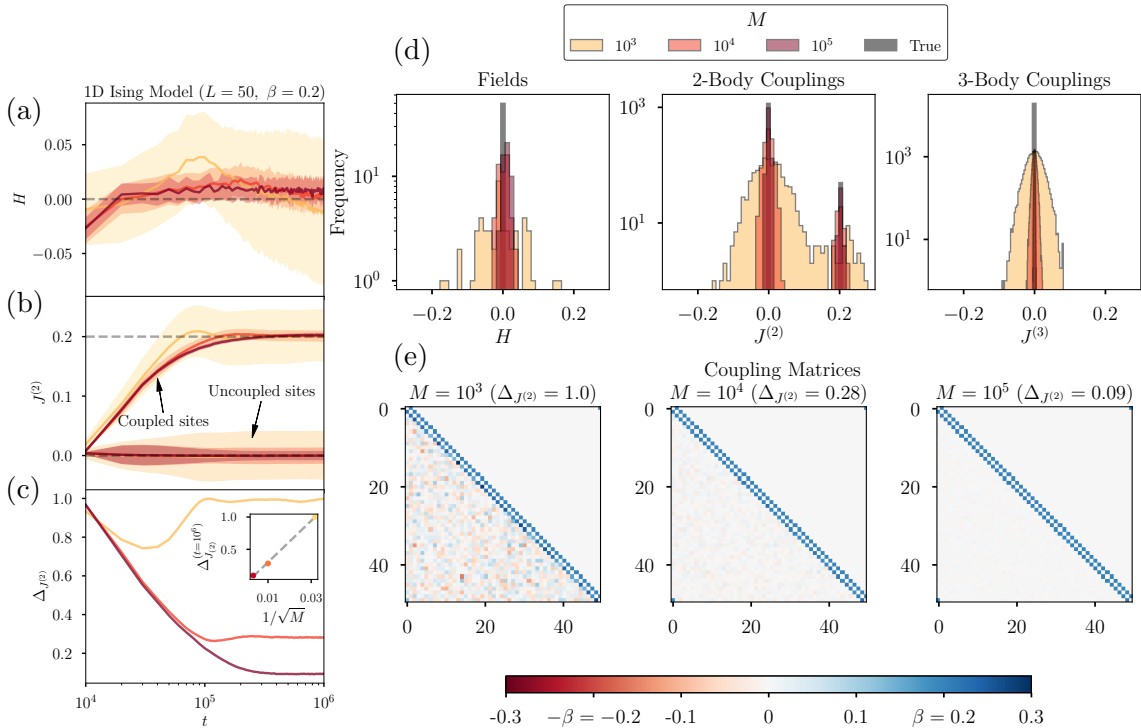

Figure 3: **RBM effective model for the ferromagnetic 1D Ising model** ($L = 50$, $\beta = 0.2$). (a) Fields, (b) pairwise couplings, (c) the error of the pairwise couplings inferred by the RBM as a function of training time $t$, given in model parameters' update units. In (a) and (b), the solid lines represent the mean of the derived parameters, while the width of the shaded area indicates their standard deviation. In (c), we add an inset showing the inference error as a function of the inverse square root of the training dataset size $M$ at $t = 10^6$. (d) Histogram of the inferred fields, pairwise and 3-body couplings and (e) inferred coupling 2-body constants matrices at the end of the training ($t = 10^6$). Note in (e) that the RBM-inferred couplings are given in the lower part of the matrices, while the upper part contains the couplings of the ground truth model. The RBMs showed were trained using the PCD-50 scheme, with $N_{\mathrm{h}} = 100$ and $\gamma = 0.01$.

the correct thermalization of the simulations and statistical independence of the samples, are given in Appendix F. In the 1D and 2D cases, we systematically compare that the expected equilibrium values for the magnetization, energy, and susceptibility are compatible with the theoretical values for finite $N$. Such tests can be found in [29].

## 3.1   Inferring the couplings of the 1D Ising Model

We considered a dataset generated by the 1D-dimensional ferromagnetic Ising chain with nearest-neighbor interactions ($J_{ij} \neq 0$ if spins $i, j$ are nearest neighbors, 0 otherwise), without external magnetic field (therefore all elements are zero for both $\boldsymbol{H}$ and $\boldsymbol{J}^{(3)}$), under periodic boundary conditions ($\sigma_{N_{\mathrm{v}}+1} \equiv \sigma_1$). We consider $N = L = 50$, $\beta = 0.2$ and $J = 1$. We perform 3 RBM trainings with the PCD-50 scheme and $\gamma = 0.1$ of an RBM with $N_{\mathrm{h}} = 100$ hidden nodes, where each training differs only in the size of the training set $M$ (here we consider $M = 10^3, 10^4,$ and $10^5$).

    The results of using our method to recover the parameters of the Ising model are shown in Fig. 3, where the different colors are associated with the different $M$s. In (a)-(c), we

evaluate the quality of inference as a function of training time, the number of gradient updates. In (a) we show the averaged value of the effective fields (which are zero in the truth model). In (b) we show the averaged value of the effective pairwise couplings, averaged over the interacting pairs in the original model, and over the non-interacting pairs of spins, separately. In both cases, the average values settle around the expected value as training progresses (highlighted by a dashed gray line). In (c), we show the error (17) in extracting 2-body couplings using the formula of Eq. (13).

It is interesting to observe the development of a minimum at short training times in the error curves when $M = 10^3$ (and to a lesser extent at $10^4$), and could be misidentified as the sweet point of inference, i.e., an *early stopping* of the training time to obtain the best inference performance. However, this minimum is just an artifact of the initialization of the RBM model. The initial weights of the matrix $\boldsymbol{W}$ follow a Gaussian distribution with center zero and low variance ($\sim 10^{-4}$), which means that all effective couplings of the RBM in the early stages of training are also distributed close to zero with low variance. Thus, at these stages, the error contribution of the couplings from uncoupled sites is very small, while the contribution of the coupled sites is very high, but they are much less numerous (the 1D Ising model is very sparse, the number of non-zero links scale as $\sim \mathcal{O}(N)$ while the total number of pairs is $\sim \mathcal{O}(N^2)$). In other words, the RBM has learned the correct connectivity matrix at the minimum, but not yet the correct temperature of the data. At the beginning of training, the error from the inferred couplings of the coupled sites decreases when the value of these couplings starts to deviate from zero, which means that the total error $\Delta_{J_{(2)}}$ decreases. However, if the training dataset is not large enough, at some point during training the error coming from effective noninteracting couplings increases toward nonzero values as RBM learning also adjusts the noise in the dataset, which should decrease with $1/\sqrt{M}$. This mixed effect between overfitting and zero initialization leads us to the existence of such a minimum when dealing with small datasets. To illustrate the role of overfitting in the final estimates of $\boldsymbol{J}$, i.e., when the training has already converged to a stationary value, we show the final value of $\Delta_{J_{(2)}}$ as a function of $1/\sqrt{M}$, which shows the expected linear scaling.

Finally, we show the histograms of the fields and the elements of the coupling matrix and of the 3-body coupling tensor in (d). We clearly see that an RBM trained with $M \sim 10^4$ or more is able to separate the non-zero couplings from the zero couplings and to infer that no 3-body coupling is present. The former can be seen in (e), where the effective coupling matrix extracted by the RBM is compared to the ground truth in the upper diagonal triangle.

It is necessary to compare RBM-inferred couplings with other standard inference methods that are better suited to the specific inverse Ising problem. The strength of RBM lies in its ability to catch also the presence of high-order interactions in complex datasets. One could think that this large flexibility could be a handicap when trying to infer correctly pairwise couplings, which are in general the most interpretable interactions in common applications. We already showed in Fig. 3 that the RBM learns to correctly identify that high-order interactions were not important for the 1D Ising data, but we can also see in Fig. 4 that the quality of the pairwise interactions inferred, and their variability, are comparable to that obtained with other approaches that only consider this level of interaction, e.g., the Boltzmann machine (BM). Aditionally, one can also extract the couplings using the Belief Propagation (BP) [30–32] formula, which becomes exact in this case as $M$ goes to infinity since the 1D Ising model have a tree-like structure locally. Still, at the temperatures studied here, the correlation is sufficiently weak to efficiently take these approximations as exact even for the small sizes we consider. In Fig. 4 we compare the $\boldsymbol{J}^{(2)}$ derived from the RBM with those inferred by training a BM or by using the BP

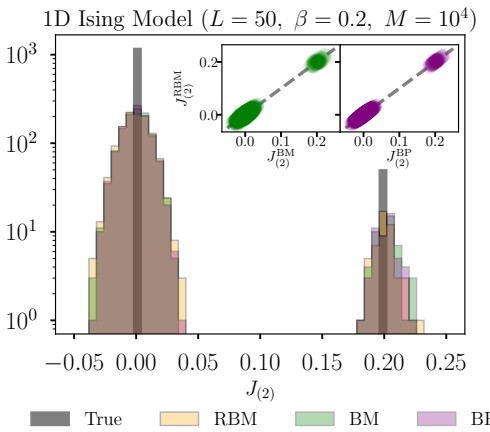

Figure 4: We compare the histogram of the pairwise couplings inferred by the RBM with those obtained using Belief Propagation (exact in the 1D Ising model in the limit of $M, N \to \infty$) and with an Inverse Ising model (a Boltzmann Machine). Showing a remarkable agreement both in the temperature inferred, and on the width of the peaks. The data set size in these examples was set to $M = 10^4$. The RBM showed were trained using the PCD-50 scheme, with $N_{\mathrm{h}} = 100$. The learning rate was $\gamma = 0.01$ both for the RBM and the BM.

formula. Our results show perfect agreement with the other methods for data generated at $\beta = 0.2$ and $M = 10^4$ both in the connectivity matrix and in recovering the correct $\beta$. We will see later that the quality of agreement between the methods depends on $M$, but also on $\beta$, mainly because at low temperatures it is more difficult to obtain good equilibrium-trained RBMs, which is detrimental to the quality of the extracted effective model. Indeed, as temperature decreases, mixing times increase and reach a point where the 50 MCMC steps used to estimate the gradient become just too short to ensure (or even reasonably approximate) the convergence of the Markov chains, even when using the PCD recipe that tends to facilitate equilibration.

In describing Fig. 3(c), we discussed that the inferred couplings decrease and eventually stabilize as training progresses, evidencing that the RBM has learned the correct model for the finite $M$ distribution of the data and sticks with it from that moment on. However, this performance is not always like this. In Fig. 5 we show the same type of analysis, but in each figure we learn datasets generated with a different $\beta$. It is clear that the lower the temperature (the higher $\beta$), the more unstable the training becomes, especially if the sample size is not large. Despite the lack of convergence of the training process to a good parameter set, it is interesting to point out that one can still develop a method to find a training epoch for which the inference of the parameters is near optimal (i.e., when $\Delta J^{(2)}$ is minimal). To find the point at which to stop learning when one does not know the actual model, it is possible to look at the maximum of the log-likelihood $\mathcal{L}$ of the test set. The log-likelihood cannot be calculated exactly because it depends on the partition function, which is intractable, but it can be approximated using Annealing Importance Sampling (AIS) [33,34]. The calculation of $\Delta J$ or this $\mathcal{L}$ is relatively too slow to compute them online during training. Instead, we store the RBM parameters at different training times evenly spaced in logarithmic scale, and analyze them offline. The cost of calculating $\mathcal{L}$ and the RBM effective parameters are discussed in Appendix G.

Indeed, the cases with $\beta = 0.4$ and $\beta = 0.8$ with only $M = 10^3$ samples are quite revealing. The log-likelihood evaluated on the training set increases with the number of gradient updates, while the quality of the inferred coupling eventually deteriorates.

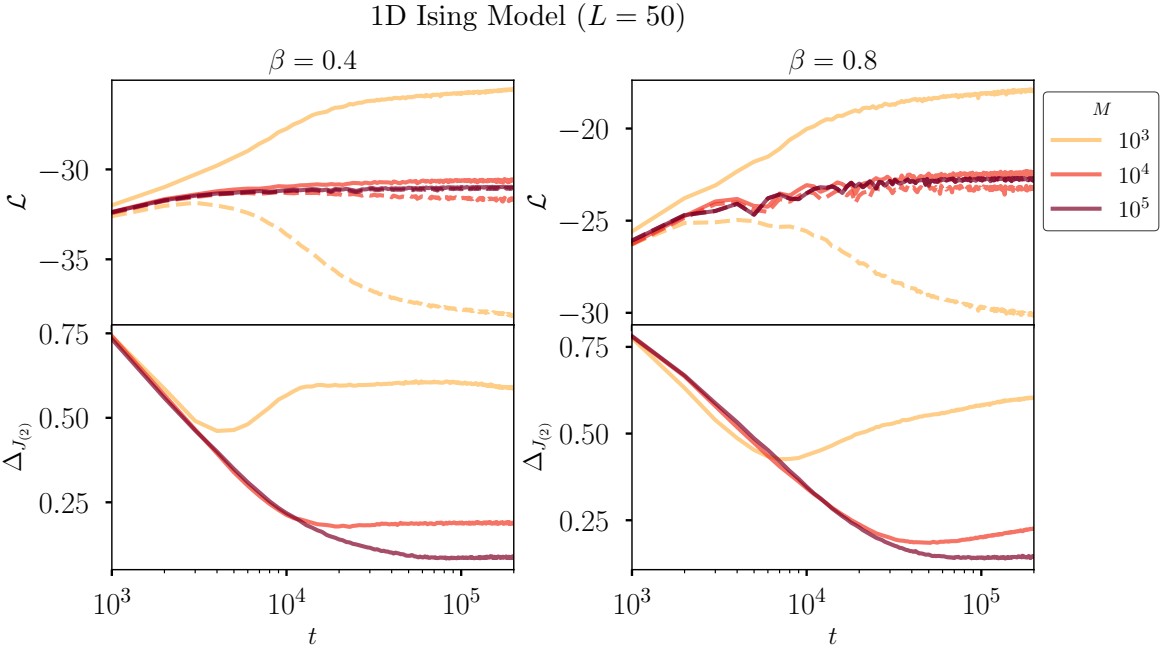

Figure 5: For data generated with the ferromagnetic 1D Ising model without external field at two different temperatures ($\beta = 1/T = 0.4$ on the left and $\beta = 0.8$ on the right), we train RBMs with three different sizes of the training set ($M = 10^3, 10^4$, and $10^5$). The colors are determined by these $M$. **Top:** we show the log-likelihood obtained with AIS [33] as a function of training time $t$. In the solid lines we show the log-likelihood of the training set given the model and in the dashed lines we show the log-likelihood for the test set. **Bottom:** Error on the RBM-inferred pairwise couplings, $\Delta_{J^{(2)}}$ defined in Eq. 17. These machines were trained using the PCD-50 scheme, with $N_{\mathrm{h}} = 100$ and $\gamma = 0.1$.

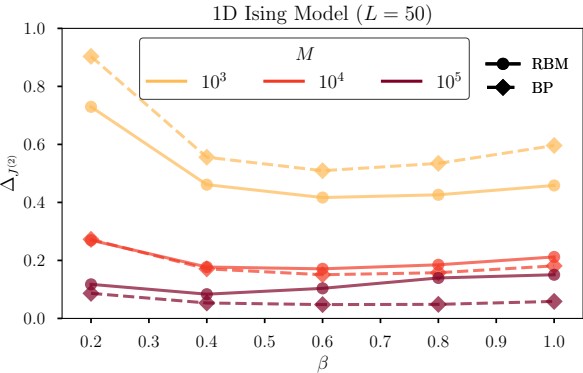

Figure 6: We show the error in the pairwise couplings, i.e. $\Delta_{J^{(2)}}$ in Eq. (17), committed by the RBM approach (in solid lines) and by the BP formula (in dashed lines) as a function of $\beta$ for the 1D Ising model. Different colors refer to different dataset sizes, $M$. The inference by BP is exact in the limiting case of $M \to \infty$ because the interactions in the 1D Ising model are tree-like. These machines were trained using the PCD-50 scheme, with $N_{\mathrm{h}} = 100$ and $\gamma = 0.1$.

However, when we look at the log-likelihood on the test set, we can clearly see that we are overfitting the RBM on the training data set. Training should stop when the

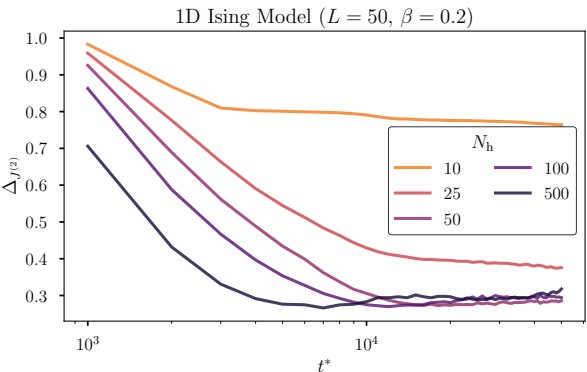

Figure 7: Error in the pairwise couplings as a function of the training time $t$ for the 1D Ising model with $\beta = 0.2$. Different colors refer to RBMs having a different number of hidden variables. These machines were trained using the PCD-50 scheme, and $\gamma = 0.1$. The size of the training set was $M = 10^4$.

log-likelihood computed on the test dataset is maximum. In some more complex cases, the lack of convergence of the inferred parameters appears (at least partially) related to problems encountered during the training process and, in particular, related to the lack of convergence of the MCMC chains used to calculate the gradient during training. It is known that in such a case EBMs learn to encode a dynamic path to reproduce the distribution of the dataset instead of encoding it on the equilibrium measure of the model [35, 36]. Furthermore, out-of-equilibrium training has been observed to fit models with pathological dynamic effects, such as incredibly slow relaxations. We will discuss these effects in detail in section 3.6.

To evaluate the quality of the RBM-derived model at different temperatures, we show in Fig. 6 the value of $\Delta_{J_{(2)}}$ at the minimum of the curves in Fig. 5 as a function of $\beta$. We compare these results with the couplings obtained using BP. As expected, our inference is worse for the largest $M$ case, where the finite $M$ corrections to the BP formula are small. But in the opposite case, when $M$ is small, the RBM performs significantly better.

Finally, we discuss the quality of inference as a function of the number of hidden nodes used to construct the RBM. As explained above, the hidden or latent variables are used to catch the correlations present in the database, and for any given dataset it is not clear how to choose this number. For the special case where the data is generated by a GIM itself, we show in Appendix D, that there is an exact construction that connects a GIM to an specific very sparse RBM that has exactly the same number of hidden nodes as the number of non-zero interactions in the model. This tells us that in our case we need at least $N_h = L = 50$ to have a model powerful enough to capture all relevant correlations in the data. As we show in Fig. 7, the final error in the inferred couplings using RBMs with $N_h$ hidden nodes decreases with $N_h$ up to 50, from then on it stays at the same value or even gets worse if the number of hidden nodes is too high, which is probably related to overfitting phenomena. The higher $N_h$ is, the faster the learning process is, since increasing $N_h$ effectively increases the learning rate.

## 3.2   1D Ising model with 3-body couplings

So far, we have only considered data generated only with a pairwise model. In this section, we generalise the previous model (the 1D Ising model) by adding also some 3-body couplings. In particular, we consider a chain of $N = 51$ Ising spins with nearest neighbor pairwise interactions as before, and add 3-body couplings (i.e. $J_{ijk}^{(3)} = 1$) when

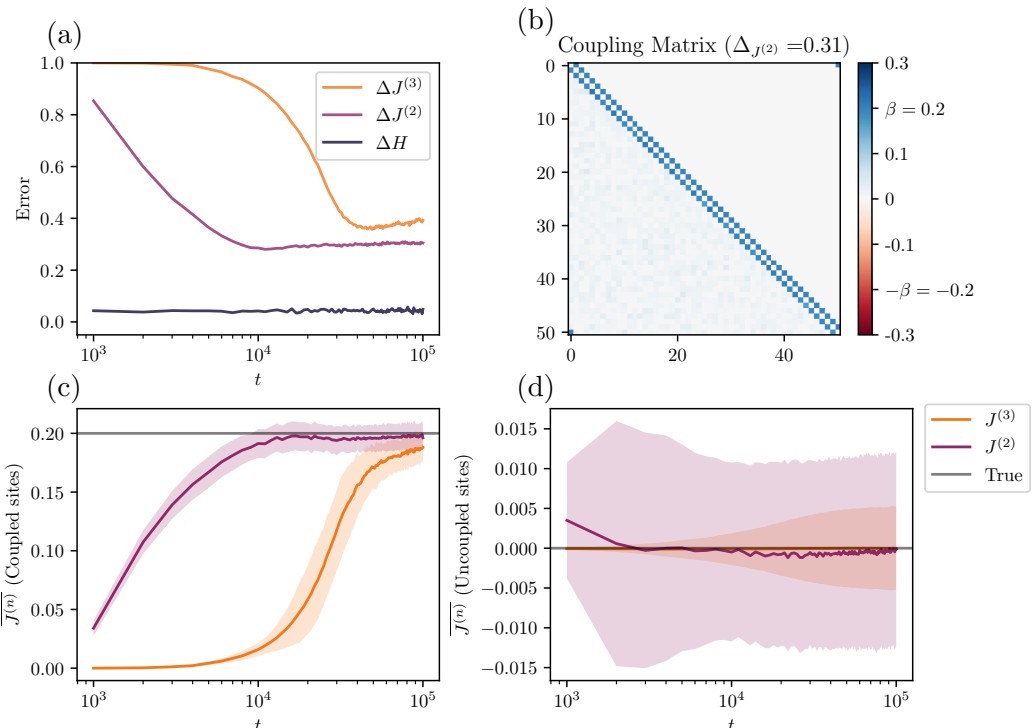

Figure 8: **Inference of 3-body couplings. (a)** We show the error in the fields, the 2- and 3-body couplings as a function of the training epochs for the 1D Ising model with 3-body interactions (see description in the text) at $\beta = 0.2$. In **(b)** we show the effective pairwise coupling matrix compared to the true matrix below and above the diagonal, respectively. In **(c)** and **(d)** we show the evolution of the average of the couplings that were non-zero and zero in the original GIM, separately, as a function of the training epochs. The shading around the lines corresponds to the standard deviation of the couplings (not the error of the mean). The machines we show here were trained using the PCD-50 scheme, with $\gamma = 0.1$, and $M = 10^4$.

$i, j$ and $k$ are subsequently separated by 17 spins each. As before, we keep the external fields at zero and generate a dataset of equilibrium configurations of this model at $\beta = 0.2$ before using it to train an RBM.

In Fig. 8 (a), we show the evolution of the errors of the inferred fields and the 2- and 3-body couplings as a function of the training time. This shows that our RBM can also correctly identify the 3-body interactions. Since the current interaction network is very sparse, we separately compare the mean value of the couplings corresponding to real bonds in the original GIM model in Fig. 8–(c) and in Fig. 8–(d) we examine the mean inferred value for the non-conected couples and triplets. This shows that the RBM not only learns the correct connectivity network, but also the correct value of $\beta$. In Fig. 8–(b), we compare the pairwise coupling matrix of the effective model with that of the original model, showing that the addition of the 3-body couplings in the model does not seem to degradate the inference of the pairwise interactions. Fig. 8–(c) shows that during training the RBM first learns to encode the 2-body couplings and in a second step the 3-body couplings.

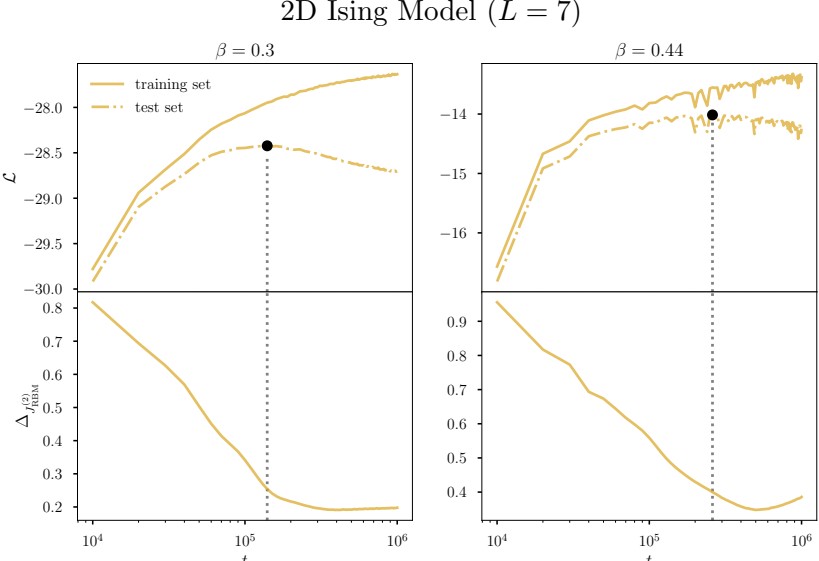

Figure 9: **Ising 2D.** Inference during training for data generated with 2D Ising model at $\beta = 0.3$ **(left)** and $\beta = 0.44$ **(right)**. As in figure 5, we show in **top** the log-likelihood $\mathcal{L}$ obtained with AIS [33] as a function of training time $t$ and in **bottom**, the error on the RBM-inferred pairwise couplings ($\Delta_{J^{(2)}}$). The vertical dotted-line is a visual guide to show the error on the coupling given by the maximum of the log-likelihood computed on the test set. These machines were trained using the PCD-50 scheme, with $N_{\mathrm{h}} = 100$ $\gamma = 0.01$ and $M = 10^4$.

## 3.3 Inferring the couplings of the 2D Ising Model

In this section we repeat the same study as in the 1D Ising case without 3-body interactions, but this time we use equilibrium configurations of the 2D Ising model with $N = 49$ spins ($L = 7$) as training set. This means that the spins lie now on a square lattice and interact with their immediate neighbors on the other sides of the four edges connecting each node. Again, $J_{ij} = 1$ if the spins $i, j$ interact, 0 otherwise. This case is generally more difficult and interesting than the 1D Ising case because the connectivity network is no longer locally tree like and the mean field approximations are only valid for high temperatures, but also because the 2D Ising model has a critical/second-order phase transition at $\beta_{\mathrm{c}} \sim 0.44$ from a paragmanetic phase at high temperatures to a ferromagnetic phase at low temperatures.

The reasons why the presence of a phase transition can affect the quality of the inference are twofold. Inference is much more difficult in the low-temperature phase (ferromagnetic) than in the high-temperature phase (paramagnetic), and so is the implementation of a high-quality training process. The difficulty in training is related to the extremely slow relaxation dynamics at and below the critical temperature. Slow dynamics means long MCMC mixing times, which means that even for the PCD training method it is a challenge to obtain reliable estimates for the log-likelihood gradient during training. This lack of thermalization of the parallel chains is detrimental to the quality of the trained machines, as discussed in Ref. [37].

The results are very similar to those we discussed for the 1D case. In Fig. 9 we show above the log-likelihood $\mathcal{L}$ of the training and test sets as a function of training time and below the error in the RBM inferred couplings $\Delta_{J^{(2)}}$ for paramagnetic configurations generated at $\beta = 0.3$ and for configurations near the critical point, at $\beta = 0.44$. Again,

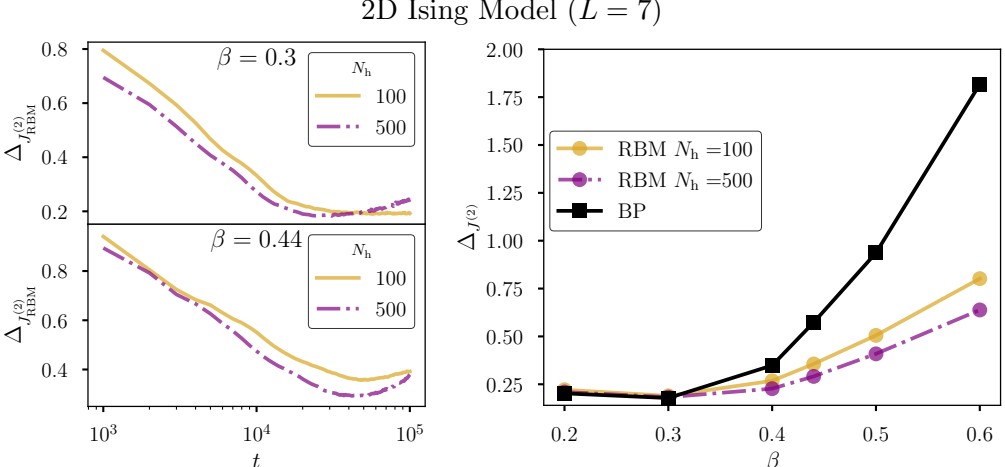

Figure 10: **Ising 2D as a function of temperature. (Left)** We show the error in the inference of the pairwise 2D Ising couplings as a function of training time for two different temperatures and two different numbers of hidden nodes $N_{\rm h}$, for $\gamma = 0.1$, i.e., 10 times faster than in Fig. 9. **(Right)** Error in the pairwise coupling matrix as a function of $\beta$ for the 2D Ising model (calculated at the minimum of the curves on the left for each $\beta$). The black square dotted lines are obtained using the Belief Propagation (BP) formula, which is no longer exact in the 2D case at $M \to \infty$. Different colors of the circular dotted lines refer to RBMs with different number of hidden variables.

the maximum of $\mathcal{L}$ computed on the test set gives a good indication of where the learning process of the RBM should stop (keep in mind that for a problem of interest one does not have access to the true generating model and thus to $\Delta_{J^{(2)}}$), while $\mathcal{L}$ evaluated on the training set shows the typical overfitting behavior that gets slower and slower as learning progresses. The test set $\mathcal{L}$ also seems to be a good indicator for the other experiments considered in this paper, but its computation was quite unstable when training at high learning rates.

The extracted coupling matrix is shown in Fig. 2 at the bottom center. As in the 1D case, the training quickly converges to a good inferred model and remains there. This is true for high temperatures and $N_{\rm h} = 100$, but at lower temperatures and a higher number of hidden nodes, the inference quality has a best training time point (i.e. it has a minimum). In Fig. 10, we compare the evolution of $\Delta_{J^{(2)}}$ during training using data obtained at $\beta = 0.3$ and $\beta = 0.44$ using RBMs with different numbers of hidden nodes $N_{\rm h} = 100$ and 500. The yellow curves ($N_{\rm h} = 100$) are analogous to those of Fig. 9-below, with the only difference being that they were obtained with a 10 times faster training ($\gamma = 0.1$ instead of $\gamma = 0.01$). The similarity between the errors obtained with the two independent trainings highlights the stability of our inference approach. On the right side, see how the reconstruction error $\Delta_{J^{(2)}}$ (at the best training time) changes when varying the inverse temperature of the generated samples. These results show that increasing the number of hidden nodes in the low temperature region proves useful, although $\sim 98$ hidden nodes should be sufficient to encode all the couplings of the model. They also clearly show that it is much better than elaborate mean-field methods (the BP approximation).

The estimated parameters obtained with our method are significantly better than those provided by the previous method proposed by Cossu et al. [15] as we will discuss in Section 3.5.

### 3.4 Inferring disordered and continuous models

We consider now consider samples generated with the 2D Edwards Anderson (EA) model [38] at zero external field, which is essentially the 2D Ising model described above, but where active couplings are $\pm 1$ with probability 50%. Unlike the previous case, there is no phase transition at finite temperature in two dimensions in the EA model, but when the temperature of the system is lowered, it still exhibits extremely slow dynamics. We use equilibrium configurations at $\beta = 0.2$ to train the RBM and analyze the most trained machine. In Fig. 11 we show the histograms of the RBM-extracted fields and pairwise couplings, as well as the matrices we obtained with different training sizes $M$ (below the diagonal the RBM-inferred ones, above the original ones). The results are consistent with our previous tests: As the number of samples increases, the inference improves and we can perfectly recover both the topology of the network and the value of the coupling constants.

So far, all generative Ising models we have considered had low-dimensional interaction geometries, in particular they were defined on regular lattices, and all active couplings had the same absolute value. We now consider a disordered model where the connectivity between the spins is defined on an Erdős–Rényi random graph with mean connectivity 4, and we allow the strength of the active couplings to fluctuate between continuous values. In particular, each active coupling is randomly chosen as plus or minus $J$, where $J$ is

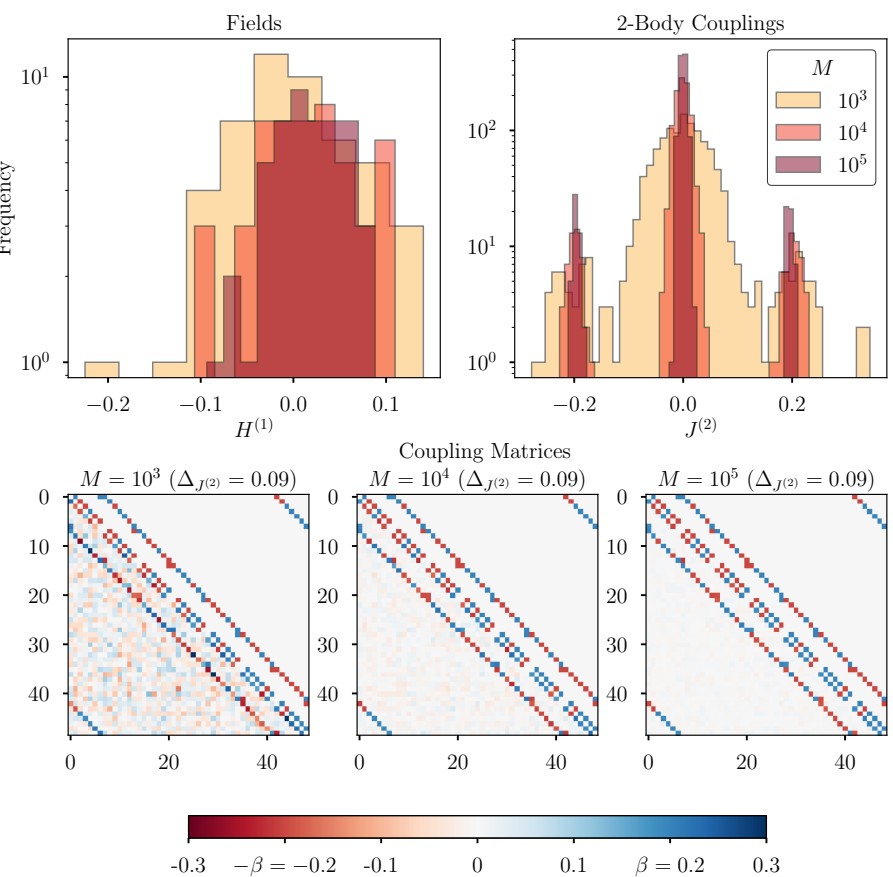

Figure 11: **2D Edwards-Anderson model.** RBM effective model for the 2D Edwards-Anderson model ($\beta = 0.2$, $L = 7$). **Top:** Histogram of the inferred fields and pairwise couplings for different sizes of the dataset, $M$. **Bottom:** Comparison between the RBM-inferred pairwise coupling matrices (below the diagonal) and the true ones (above) for different $M$.

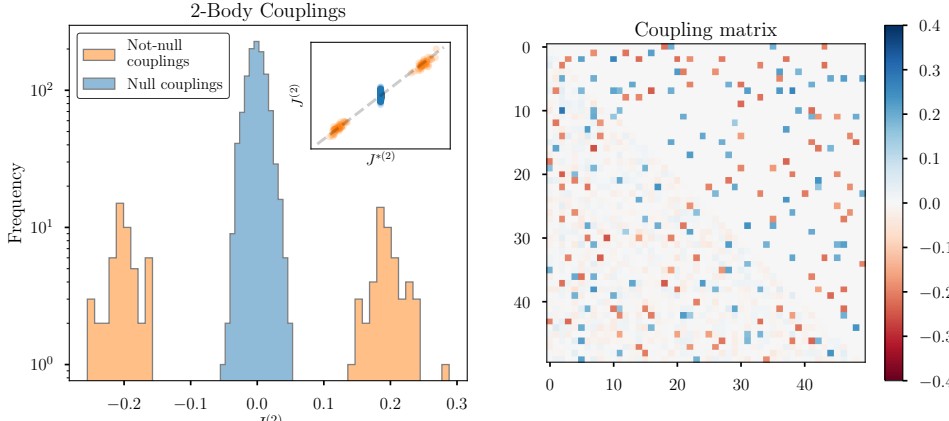

Figure 12: **Continuous couplings on a random graph.** RBM effective model for an Ising model on a random graph with random disordered and continous interactions ($N = 50, \beta = 0.2$). **(Left)** Histogram of 2-body inferred coupling constants. We plotted in different colors the couplings that in the ground truth model are null from the ones that are not null. Moreover, coupling strengths in this model were normally distributed around $-1$ or $1$ with equal probability with a scale parameter fixed to 0.1. In the inset of the left figure, we show the correlation of the RBM-inferred coupling constants $J^{(2)}$ with those of the ground truth model $J^{*(2)}$. **(Right)** The RBM-inferred couplings are given in the lower part of the matrix, while in the upper part, we show that of the ground truth model. The RBM showed here were trained under the PCD-50 scheme, with $N_{\mathrm{h}} = 100, \gamma = 0.01$ and a number of samples $M = 10^4$ .

a normal variable with mean 1 and standard deviation 0.1. We show the results of the inference process in Fig. 12, which again show very good performance.

## 3.5   Comparison with previous methods

In this subsection, we compare the quality of the inferred models obtained with our mapping with those obtained with the formulas previously proposed by Cossu *et al.* [15] for the mapping between the RBM and the $\pm 1$ Ising model, and with those obtained for the RBM- $\{0, 1\}$ lattice gas mapping proposed by Bulso *et al.* [16].

### 3.5.1   Comparison with Cossu *et al.*

In Appendix C we analytically compute the error of the expression proposed in Ref. [15] to infer the pairwise couplings of Ising-like models. It is interesting to investigate numerically how the neglected fluctuations affect the quality of the inferred parameters. In Fig. 13, we illustrate the difference in deriving the couplings of the 1D and 2D Ising models with the two methods using the same trained RBM. We clearly see that our equations are qualitatively more accurate, both in identifying the topology of the network and the strength of the couplings $\beta$, as well as in reducing the variance of the inferred couplings.

### 3.5.2   Comparison with Bulso *et al.*: Inferring the couplings of a 1D Lattice Gas Model

The mapping between the RBM and a generalized $\{0, 1\}$ lattice gas proposed by Bulso *et al.* in Ref. [16] is mathematically correct, but as mentioned above, it is rather unreliable in

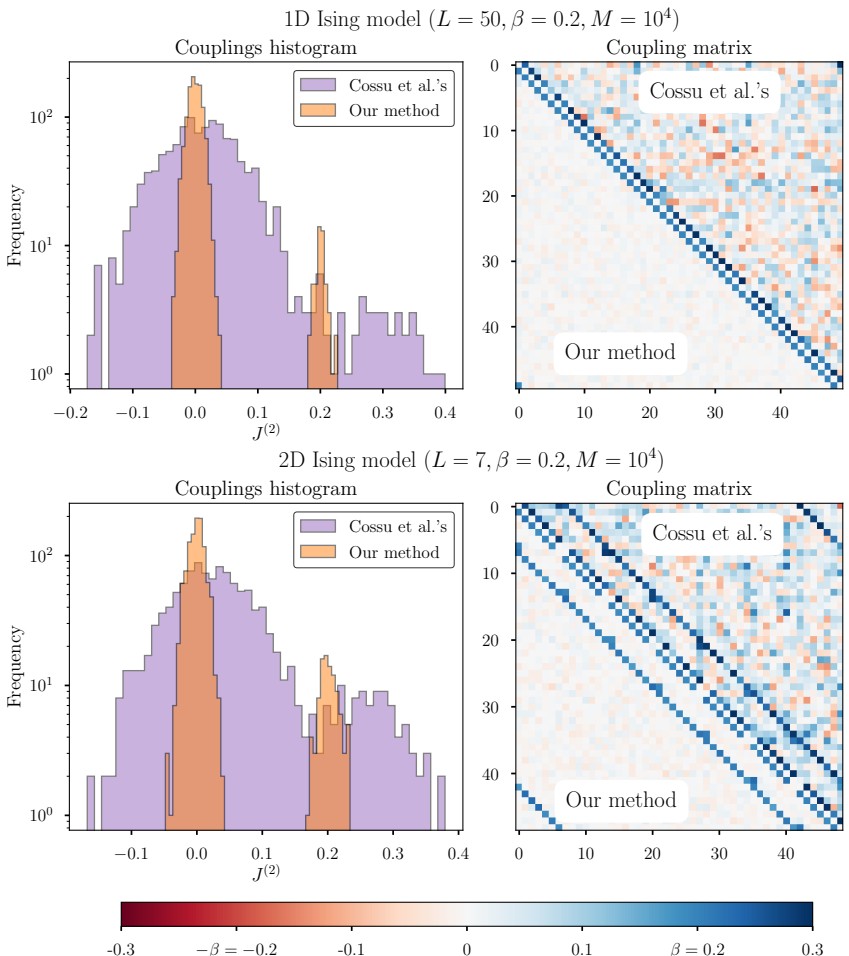

Figure 13: **Comparison with the mappin proposed by Cossu *et al.* Ref. [15].** **(Left)** We show the histograms of the couplings for the inferred 2-body couplings with both methods. **(Right)** Coupling matrices for the inferred effective models. Below the diagonal, we show the matrix inferred with our method, and above, the corresponding coupling matrix calculated with the formula obtained in [15].

practice for inverse experiments. In this section, we illustrate its problems by applying our method to determine the parameters of a 1D lattice gas model, a problem where the $\{0,1\}$ mapping should work perfectly, but still yields very inaccurate inferred model parameters.

We now use the following 1D lattice gas Hamiltonian, i.e,

$$\mathcal{H}_{\mathcal{D}}(\boldsymbol{v}) = -\sum_{j=1}^{L} v_j v_{j+1}, \text{ with } v_j \in \{0,1\}, \tag{18}$$

under periodic boundary conditions, we set $v_{L+1} \equiv v_1$, to generate equilibrium samples at $\beta = 0.8$. Then we train an RBM with this data and use the formulas from Ref. [16] (Eq. (35) in Appendix B) to infer the original lattice gas model of Eq. (18) (having zero external fields, zero pairwise interactions except for next neighboring sites $(i, i+1)$, and zero 3-body interactions). In Fig. 14–top we show the histograms of the inferred external fields, Fig. 14–(a1), pairwise couplings in Fig. 14–(b1) and in Fig. 14–(b3) the derived 3-body interactions obtained with the mapping of Ref. [16]. For the pairwise couplings, we

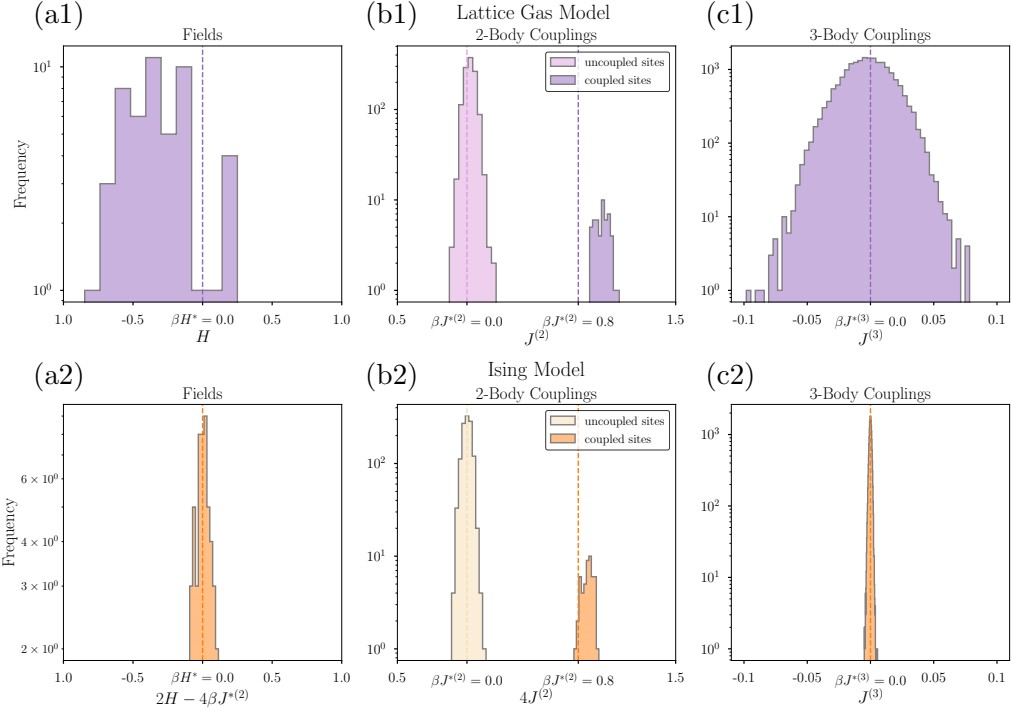

Figure 14: **Limitations of the lattice-gas mapping.** Histograms for the effective model parameters inferred with an RBM trained with equilibrium samples of the 1D lattice gas model, defined in Eq. (18) (for $L = 50$, $\beta = 0.8$). In **(a)** we show the fields, in **(b)** the pairwise couplings and in **(c)** the 3-body couplings. In the top row **(1)** we show the data inferred using the mapping between the RBM and a generalized lattice gas model (formulated in terms of Bernoulli $\{0, 1\}$ variables) proposed in Refs. [15, 16]. In the bottom row **(2)**, we show the parameters obtained using our equations (12)-(14) mapped to the lattice gas case. In all figures, the dashed vertical lines indicate the values of the parameters in the ground truth model. For the pairwise coupling inference, we show separately the histograms of the couplings corresponding to the existing and non-existing couplings in the original model. The RBMs shown were trained with the PCD-50 scheme, with $N_{\rm h} = 100$ and $\gamma = 0.01$ after $t = 10^6$ parameter updates. The size of the training dataset in this case was $M = 10^5$.

separately compute the histogram for the couplings that were zero in the original model and those that were $\beta$ in a darker shade to emphasize that the mapping correctly identifies the right connectivity matrix, i.e., the pair of spins that interact. However, we see that the mapping does not correctly infer the strength of the interaction, i.e. $\beta$. We also see that the mapping fails to infer correctly the external null fields and predicts some rather large 3-body interactions.

One can easily rewrite the previous lattice gas model in Eq. (18) in terms of Ising spin variables $\sigma_i \in \{-1, 1\}$ by using the transformation (5) as:

$$\mathcal{H}_{\mathcal{D}}(\boldsymbol{\sigma}) = -\frac{1}{4} \sum_{j=1}^{L} \sigma_j \sigma_{j+1} - \frac{1}{2} \sum_j \sigma_j. \tag{19}$$

This means that we can then use our mapping in Eqs. (12)-(14) to obtain the effective parameters of this Ising model, that is, trying to recover $H_i = \beta/2$ for all spins, $J_{ij}^{(2)} = \beta/4$

if $j = i \pm 1$, zero otherwise, and $J^{(3)}_{ijk} = 0$ for all triplets $(i, j, k)$ in the system. We show the histogram of the derived parameters in Fig. 14–bottom, after translating them to the original scale of the lattice gas model. We see that our mapping is much more accurate than previous attempts, even when it comes to the case where the mapping between the RBM and the lattice gas should be perfect.

### 3.6 Out-of-equilibrium training effects

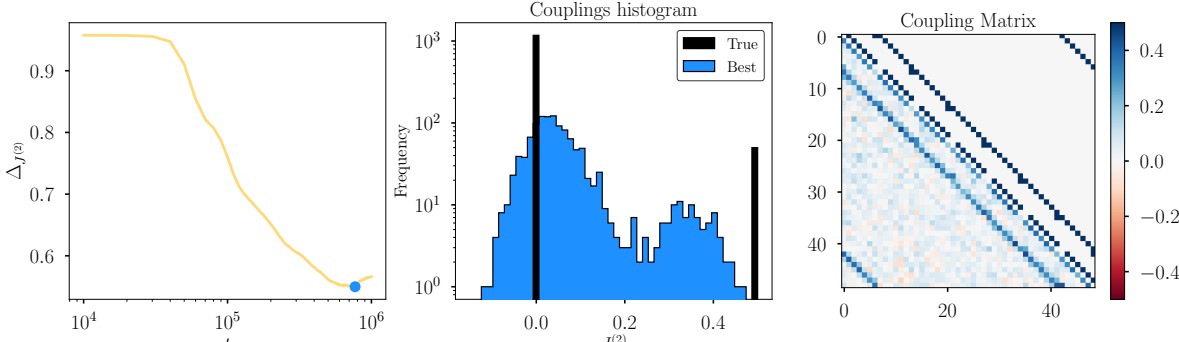

Figure 15: **Inference of the 2D Ising model using an out-of-equilibrium training (Rdm-10 training scheme).** We consider $N = 7^2$ spins, $\beta = 0.5$ and $M = 10^4$. From left to right: (i) the error in reconstructing the 2-body couplings as a function of training time; (ii) the histogram of couplings obtained at the training time, indicated by the blue dot; (iii) the reconstructed coupling matrix at that time. We see that the inference in this case is quite poor, probably aggravated by the fact that the samples are taken in the ferromagnetic phase.

As other EBMs, RBMs need Markov Chain Monte Carlo (MCMC) both for generating samples, and to estimate the log-likelihood gradient during the training. Recently it has been shown that the lack of thermalization of these runs during the training my introduce memory effects into the RBM [24, 35] that should affect the quality of the effective model we extract, as described recently for the Boltzmann machine [36].

To investigate how an out-of-equilibrium training affects the reliability of the effective models, we train the RBMs following a purely out-of-equilibrium strategy (namely Rdm-10 scheme): each time we want to estimate the gradient during learning, we initialize the chains randomly and perform only 10 MCMS steps before computing the estimated values. Despite the obvious flaw of this method to produce equilibrated samples, such a training procedure is able to produce data of very good quality [35, 39] even for very structured datasets where the standard PCD approach cannot provide reasonable models [37]. This is also the case when learning an RBM with the 2D Ising equilibrium samples. Indeed, one can easily verify that samples drawn with a trained model obtained by performing 10 MCMC steps from a random initialization (the same procedure used for training) perfectly describe the equilibrium finite-size statistics of energy, magnetization, susceptibility, and specific heat [29]. It is then an interesting question whether such RBMs provide reasonable effective coupling matrices or not.

To this end, instead of using the standard PCD training scheme used all over the experiments in this paper, we train a new RBM with 2D Ising data close below the critical point $\beta = 0.5$, following this out-of-equilibrium scheme. After the training, we apply our mapping to extract the effective couplings and fields. We show the results of the

analysis in Fig. 15. First, we show the evolution of the error in the pairwise couplings as a function of training updates. Typically we see that OOE training can quickly lead to very poor equilibrium models when training times are long, which is notably reflected in the log-likelihood [35]. For the best machine, highlighted with a blue dot, we show the unnormalized histogram of the inferred coupling matrix elements. We see that the OOE training tends to infer weak next-to-nearest neighbors interactions that are not present in the true model with values of $J_{ij}^{(2)}$ just slightly above zero.

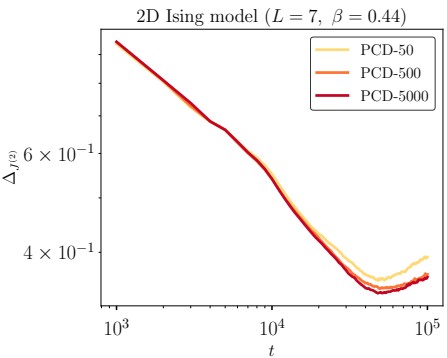

Figure 16: **PCD-$k$ effects.** Error in the pairwise coupling matrix as a function of training time $t$ for the 1D (**left**) and 2D (**right**) Ising models. Different colors refer to different number of Gibbs steps $K$ used to compute the gradient. For the 1D Ising case, we also show results for different dataset sizes $M$.

The effects of the lack of thermalisation when using the standard PCD training procedure are harder to evaluate, because the PCD is designed to try to enhance thermalisation at much as posible. In this scheme, a permanent chain of spins is kept all along the training, which means that samples are slowly annealed (as the parameters are slowly changed). Nearby the critical point $\beta = 0.44$, one expects still to observe some out-of-equilibrium effects in the model because relaxation times diverge. It is then reasonable to thing that the longer we iterate the MCMC process during the training, the better it will be the equilibration. With this aim, we show in Fig. 16 the evolution of the error in reconstructing the pairwise couplings as a function of training time for 3 different training procedures in which we iterate $K$ times the MCMC process prior to each time we want to infer the model parameters. We refer to these different training processes as PCD$-K$ training, with $K = 50, 500$ and $5000$. In general, we see that the longer $K$, the better the reconstruction, although the effect is modest.

# 4 Limitations of the current approach

The RBM inference method presented in the paper has several limitations worth to mention. First, the current mapping can only be applied to binary data sets, i.e. mainly to data formulated either in the form of Ising spins $\pm 1$ or Bernouilli $\{0, 1\}$. For the latter, one would first have to derive the effective Ising model and rewrite the variables to obtain the effective lattice gas model. Such an approach means ignoring the contributions of higher-order interactions to the lower-order couplings $\{0, 1\}$, but we have shown in section 3.5.2 that it is still more accurate than deriving an effective lattice gas model directly. There are many inference problems in science involving binary variables (see, e.g., [7,9–11]), but many other applications, such as contact prediction in proteins, would require more general mappings involving categorical/Potts variables or even continuous variables for signal

or image processing. The generalization of the equations (12)-(14) to different types of variables is not straightforward but feasible and will be the subject of future studies.

Second, the formula (14) allows us to extract couplings up to an arbitrarily high order $n$, but in practice, computing all $J^{(n)}_{j_1,...,j_n}$ contributions for $n = 4$ becomes tedious and quickly impossible in a reasonable time. However, we would like to emphasize that in practical situations, this allows you to explicitly compute the $n$-body interactions within the $n-$tuple group of variables for which you suspect a particular interaction for external regions, for example in regions with a particular biological function. But even if you are not interested in computing the higher-order interaction terms, your trained RBM has these terms encoded, making it a much more powerful generative model than a BM if your goal is to generate reliable synthetic data [40, 41]. These RBMs are also better models to learn from your data, i.e. they can be used to explore the properties of the landscape for practical applications, e.g. for family/subfamily detection [24] or for studying mutational pathways [42].

As a pairwise inference method, RBM is of course much slower than other direct methods such as BP, which do not require the training of a model (even though the training of the models presented in this paper was quite fast, see Appendix G). Rather, the advantage of this approach is that it provides you with a direct interpretation of a model that can be used for many other applications. In addition, the ability to encode high-order interactions improves the quality of pairwise inference for problems where these high-order terms are relevant. No less importantly, the generalization of the mappings based on an RBM is much easier than the generalization of BP formulas to deal with more complex data structures. It therefore seems to make more sense to compare the computational cost of deriving couplings with an RBM versus a BM. Training a BM using maximum likelihood is extremely slow, mainly because spin updates cannot be parallelized. In this sense, the bipartite nature of the RBM provides a natural way to update spins asynchronously, allowing for dramatic speedups of training in GPU architectures. We compare the times required to train a BM with those required to train an RBM in Appendix G. However, while extracting the coupling matrix in BMs is straightforward, after training an RBM, we need to reconstruct the effective GIM, which can take some time, as we discuss in Appendix G.

## 5 Conclusions

We have shown that RBMs combine the power of neural networks to encode the full diversity of complex datasets, while remaining as interpretable for contact prediction as Boltzmann machines or Inverse Ising approaches. We have also shown that they can be used as inference engines to extract couplings up to a potentially arbitrary order of interaction. We have tested this idea in a series of controlled experiments, in which we have shown that RBMs perform as well as the Boltzmann machine in inverse Ising tasks, but are additionally able to accurately determine the presence of third-order interactions, which also leads to better inference of pairwise interactions in these situations.

The formulas we obtained for the mapping between an RBM and a generalized Ising model are easy to implement and a corresponding Python code using them is freely available on Github. Our work paves the way for the use of these mappings in relevant scientific applications such as neuroscience or bioinformatics. Our mapping provides a simple way to generalize the standard pairwise epistatic fitness models in evolutionary genomics [9, 10] or in DCA approaches used in protein structure prediction [8]. However, the mapping developed in this work is only valid for the description of binary data sets. Further ex-

tensions of these relationships to describe other types of datasets, such as homologous protein sequences formulated in terms of Potts/categorical variables, will be tackled in future work.

We have also shown that the quality of the training protocol strongly influences the quality of the learned model used for inference tasks, even if the model itself can perform very well as a generative model. For this reason, we need to strive to find better algorithms to mitigate the introduction of unwanted out-of-equilibrium effects into the learned model.

# Acknowledgements

We thank Giovanni Catania for his help in performing maximum likelihood training with the Boltzmann Machine to compare inference results with our data. Also, we thank ECT* and the ExtreMe Matter Institute EMMI at GSI, Darmstadt, for support in the framework of the ECT*/EMMI Workshop: *Machine Learning for Lattice Field Theory and Beyond* during which discussions that increase the quality of the present work took place.

**Code availability** The python code to extract the effective couplings given a RBM can be found at https://github.com/alfonso-navas/inferring_effective_couplings_with_RBMs. The code to train Bernouilli-Bernouilli RBMs is available at https://github.com/AurelienDecelle/TorchRBM.

**Funding information** We acknowledge financial support by the Comunidad de Madrid and the Complutense University of Madrid (UCM) through the Atracción de Talento programs (Refs. 2019-T1/TIC-13298 and 2019-T1/TIC-12776), the Banco Santander and the UCM (grant PR44/21-29937), and Ministerio de Economía y Competitividad, Agencia Estatal de Investigación and Fondo Europeo de Desarrollo Regional (Ref. PID2021-125506NA-I00).

# A The Restricted Boltzmann Machine

In this section, we discuss the main features of the RBM model and its training procedure. For a comprehensive introduction to RBMs, the reader is referred to Ref. [43], and for a presentation of generalized models and a review of recent advances in statistical physics to Ref. [17].

## A.1 The RBM definition

An RBM is an undirected stochastic neural network defined on a bipartite graph consisting of two layers of variables (or units): the *visible* nodes and the *hidden* nodes. The visible layer includes $N_{\mathrm{v}}$ units $v_j$, with $j = 1, \ldots, N_{\mathrm{v}}$, and represents the configurations of the data. Also, the hidden layer contains $N_{\mathrm{h}}$ units $h_i$, with $i = 1, \ldots, N_{\mathrm{h}}$, on which the latent representations of the data are carried (as we will see, they are used to encode the correlations in the data). Here we will define both visible and hidden nodes using the binary support $\{0, 1\}$. The interactions between the nodes in the visible and hidden layers are given by the coupling matrix $\boldsymbol{W}$, whose elements are denoted by $W_{ij}$. In addition, each variable can have a local magnetic field, referred to as a *bias* in ML technical jargon. We will use $b_j$ and $c_i$ to denote the bias of the visible and hidden nodes, respectively. Thus, we can introduce the Hamiltonian or the energy function of the joint state $\{\boldsymbol{v}, \boldsymbol{h}\}$

of the machine:

$$\mathcal{H}(\boldsymbol{v}, \boldsymbol{h}) = -\sum_{i=1}^{N_\mathrm{h}} \sum_{j=1}^{N_\mathrm{v}} h_i W_{ij} v_j - \sum_{j=1}^{N_\mathrm{v}} b_j v_j - \sum_{i=1}^{N_\mathrm{h}} c_i h_i. \tag{20}$$

For a graphical representation of the RBM architecture, see Fig. 1–B1. The probability of such a given state is determined by the Gibbs-Boltzmann distribution:

$$p(\boldsymbol{v}, \boldsymbol{h}) = \frac{1}{\mathcal{Z}} e^{-\mathcal{H}(\boldsymbol{v}, \boldsymbol{h})}, \tag{21}$$

with the partition function $\mathcal{Z}$ defined as

$$\mathcal{Z} = \sum_{\boldsymbol{v}, \boldsymbol{h}} e^{-\mathcal{H}(\boldsymbol{v}, \boldsymbol{h})}, \tag{22}$$

where the sum $\sum_{\boldsymbol{v}, \boldsymbol{h}}$ runs over all possible configurations of the model. From the above expression, it is immediately clear that the Gibbs-Boltzmann distribution defined in Eq. (21) is correctly normalized.

Given the bipartite structure of the RBM, the hidden variables $\boldsymbol{h}$ are statistically independent given the state of the visible layer $\boldsymbol{v}$, and vice versa, i.e.,

$$p(\boldsymbol{h}|\boldsymbol{v}) = \prod_{i=1}^{N_\mathrm{h}} p(h_i|\boldsymbol{v}) \quad \text{and} \quad p(\boldsymbol{v}|\boldsymbol{h}) = \prod_{j=1}^{N_\mathrm{v}} p(v_j|\boldsymbol{h}). \tag{23}$$

Thus, the conditional probability of a single node can be readily computed:

$$p(h_i = 1|\boldsymbol{v}) = \sigma\left(\sum_{j=1}^{N_\mathrm{v}} W_{ij} v_j + c_i\right) \quad \text{and} \quad p(v_i = 1|\boldsymbol{v}) = \sigma\left(\sum_{i=1}^{N_\mathrm{h}} h_i W_{ij} + b_j\right), \tag{24}$$

where $\sigma(x) = (1 + e^{-x})^{-1}$ is the sigmoid function.

## A.2   The RBM training

The training of an RBM model consists of tuning the weight matrix $\boldsymbol{W}$ and the biases $\boldsymbol{b}$ and $\boldsymbol{c}$ so that the Gibbs-Boltzmann distribution defined in the Eq. (21) resembles as much as possible to the data distribution. The classical procedure to do so is by maximizing the log-likelihood (LL) function of the training dataset $\mathcal{D} = \left\{\boldsymbol{v}^{(1)}, \ldots, \boldsymbol{v}^{(M)}\right\}$ via *Gradient Ascent*. Such a LL function is defined as

$$\mathcal{L} = \frac{1}{M} \sum_{m=1}^{M} \log p(\boldsymbol{v}^{(m)}) = \frac{1}{M} \sum_{m=1}^{M} \log \sum_{\boldsymbol{h}} e^{-H(\boldsymbol{v}^{(m)}, \boldsymbol{h})} - \log \mathcal{Z}, \tag{25}$$

being its gradient given by

$$\frac{\partial \mathcal{L}}{\partial W_{ij}} = \left\langle h_i v_j \right\rangle_\mathcal{D} - \left\langle h_i v_j \right\rangle_\mathrm{h}, \quad \frac{\partial \mathcal{L}}{\partial b_j} = \left\langle v_j \right\rangle_\mathcal{D} - \left\langle v_j \right\rangle_\mathrm{h} \quad \text{and} \quad \frac{\partial \mathcal{L}}{\partial c_i} = \left\langle h_i \right\rangle_\mathcal{D} - \left\langle h_i \right\rangle_\mathrm{h}, \tag{26}$$

where we consider two different types of averages. The average over the training set is defined as

$$\left\langle f(\boldsymbol{v}, \boldsymbol{h}) \right\rangle_\mathcal{D} = \frac{1}{M} \sum_{m=1}^{M} \sum_{\boldsymbol{h}} f(\boldsymbol{v}^{(m)}, \boldsymbol{h}) p(\boldsymbol{h}|\boldsymbol{v}^{(m)}), \tag{27}$$

while, $\langle f(\boldsymbol{v}, \boldsymbol{h}) \rangle_{\mathrm{h}}$ denotes the average over the Gibbs-Boltzmann measure in Eq. (21), i.e.

$$\langle f(\boldsymbol{v}, \boldsymbol{h}) \rangle_{\mathrm{h}} = \frac{1}{\mathcal{Z}} \sum_{\boldsymbol{v}, \boldsymbol{h}} f(\boldsymbol{v}, \boldsymbol{h}) e^{-\mathcal{H}(\boldsymbol{v}, \boldsymbol{h})}. \tag{28}$$

Once the gradient in Eq. (26) is computed, the parameters of the model can be updated according to

$$W_{ij}^{(t+1)} = W_{ij}^{(t)} + \gamma \left. \frac{\partial \mathcal{L}}{\partial W_{ij}} \right|_{\{\boldsymbol{W}^{(t)}, \, \boldsymbol{b}^{(t)}, \, \boldsymbol{c}^{(t)}\}},$$

$$b_j^{(t+1)} = b_j^{(t)} + \gamma \left. \frac{\partial \mathcal{L}}{\partial b_j} \right|_{\{\boldsymbol{W}^{(t)}, \, \boldsymbol{b}^{(t)}, \, \boldsymbol{c}^{(t)}\}},$$

$$c_i^{(t+1)} = c_i^{(t)} + \gamma \left. \frac{\partial \mathcal{L}}{\partial c_i} \right|_{\{\boldsymbol{W}^{(t)}, \, \boldsymbol{b}^{(t)}, \, \boldsymbol{c}^{(t)}\}}.$$

In the above expressions, we have introduced $t$ to denote the $t$-th RBM parameter update and $\gamma$ as the *learning rate*, a parameter that controls the pace at which the RBM parameters are updated in the direction of steepest ascent.

The most difficult part of implementing the above procedure lies in the computation of the gradient of the LL function. Although the positive terms of the gradient in Eq. (26) can be calculated directly, the negative terms cannot be obtained exactly and depend on the RBM model parameters at each stage of the training. Therefore, they must be estimated from equilibrium samples of the RBM. Such samples are generated using a MCMC method called *Gibbs sampling*. For RBMs, *Alternating Gibbs sampling* consists of implementing the following Markov chain

$$\boldsymbol{v}^{(0)} \to \boldsymbol{h}^{(0)} \to \boldsymbol{v}^{(1)} \to \boldsymbol{h}^{(1)} \cdots \to \boldsymbol{v}^{(k)} \to \boldsymbol{h}^{(k)},$$

where $\boldsymbol{h}^{(l)}$ and $\boldsymbol{v}^{(l+1)}$ are sampled using the conditional probabilities given in Eqs. (23) and (24). Sampling equilibrium configurations of the RBM during gradient training estimation is critical to obtain effective models whose equilibrium distribution matches the distribution of the data [35]. Although equilibrium can be achieved by any chain with a sufficiently large number of MCMC steps $K$, the implementation of long chains is computationally expensive because samplings must be repeated each time the parameters are updated during learning. Therefore, part of the success of RBM training is to find a clever starting point $\boldsymbol{v}^{(0)}$ for the chains to reduce the number of MCMC steps required to reach equilibrium. In this context, we used the *Persistent Contrastive Divergence* (PCD-$K$) [44], which consists in setting the initial state of the chains $\boldsymbol{v}^{(0)}$ in the $t$-th RBM parameter update as the final state $\boldsymbol{v}^{(k)}$ obtained when computing the gradient in the previous parameter update, i.e. the $(t-1)$-th one. The logic behind such a procedure is based on the fact that since the parameters change smoothly during learning, the same should be expected for the equilibrium configurations of the model. Indeed, it has been shown numerically that the RBM often achieve a quasi-equilibrium [35] with this recipe, even though the convergence of the sampling procedure depends strongly on the properties of the training set.

To investigate the impact of the lack of thermalization during training on the effective models extracted from the trained RBMs, we will also consider another completely out-of-equilibrium recipe in which each chain is always initialized with random conditions at each update. This recipe was recently proposed in [35, 45] as a very efficient method for training models capable of generating high-quality synthetic samples at very short sampling times. We refer to this training recipe as Rdm-$K$.

Using the training protocol described earlier, we also implemented a *stochastic gradient ascent* approach to train our RBM models. This approach consists of dividing the training dataset into sets, called *mini-batches*, over which the gradient ascent was sequentially implemented in a random order, rather than simply updating the parameters by computing the gradient using the entire dataset at each iteration step. A cycle over all mini-batches is referred to as *epoch*, resulting in a number of parameter updates per epoch equal to the number of mini-batches.

# B Alternative derivation of the equivalence between the RBM and the generalized lattice gas model

In this section, we present an alternative derivation of the expression proposed in [15, 16, 26], which matches the energy function of the marginalized RBM with a generalized gas lattice model, i.e., a model defined as Eq. (1), but in terms of particle variables $v_j \in \{0, 1\}$ (i.e. occupied/unoccupied sites), i.e:

$$\mathcal{H}_{\mathrm{GIM}}(\boldsymbol{v}) = -\sum_j H_j v_j - \sum_{j_1 > j_2} J^{(2)}_{j_1 j_2} v_{j_1} v_{j_2} + \cdots - \sum_{j_1 > \cdots > j_n} J^{(n)}_{j_1 \ldots j_n} v_{j_1} \ldots v_{j_n} + \cdots . \quad (29)$$

In this case, let us start with the expression of the marginalized energy given in Eq. (4):

$$\mathcal{H}(\boldsymbol{v}) = -\sum_j b_j v_j - \sum_i \ln \sum_{h_i} \exp\left(c_i h_i + \sum_j h_i W_{ij} v_j\right).$$

$$= -\sum_j b_j v_j - \sum_i \ln\left(1 + e^{c_i + \sum_j W_{ij} v_j}\right). \quad (30)$$

Analogous to what was done in the Section 2.2, we introduce the auxiliary variables $v'_j \in \{0, 1\}$ and express the Kronecker delta symbol $\delta_{v_j v'_j}$, taking into account that in this case the following identity holds:

$$\delta_{v'_j, v_j} = 1 + v_j(v'_j - 1) + v'_j(v_j - 1)$$

$$= 1 - v'_j + (2v'_j - 1)v_j. \quad (31)$$

Thus, one can write Eq. (30) as

$$\mathcal{H}(\boldsymbol{v}) = -\sum_j b_j v_j - \sum_{\boldsymbol{v}'} \prod_j \delta_{v_j v'_j} \sum_i \ln\left(1 + e^{c_i + \sum_j W_{ij} v'_j}\right).$$

$$= -\sum_j b_j v_j - \sum_{\boldsymbol{v}'} \prod_j \left(1 - v'_j + (2v'_j - 1)v_j\right) \sum_i \ln\left(1 + e^{c_i + \sum_j W_{ij} v'_j}\right). \quad (32)$$

By expanding the right-hand side of the above expression and comparing it term-by-term to Eq. (29), we obtain an expression for the $n$-body coupling constant $J^{(n)}_{j_1 \ldots j_n}$:

$$J^{(n)}_{j_1 \ldots j_n} = \sum_{\boldsymbol{v}'} \left[\prod_{\mu=1}^n \left(2v'_{j_\mu} - 1\right)\right] \left[\prod_{\mu=n+1}^{N_{\mathrm{v}}} \left(1 - v'_{j_\mu}\right)\right] \sum_i \ln\left(1 + e^{c_i + \sum_j W_{ij} v'_j}\right). \quad (33)$$

Note that terms in the sum above where $v'_{j_\mu} = 1$, with $\mu > n$, cancel out. Therefore, Eq. (33) is reduced to

$$J^{(n)}_{j_1 \ldots j_n} = \sum_{v'_{j_1}} \cdots \sum_{v'_{j_n}} \prod_{\mu=1}^n \left(2v'_{j_\mu} - 1\right) \sum_i \ln\left(1 + e^{c_i + \sum_{\mu=1}^n W_{ij_\mu} v'_{j_\mu}}\right). \quad (34)$$

Finally, we can rewrite the above expression in a form that resembles more those provided in [15, 16, 26]:

$$J_{j_1 \dots j_n}^{(n)} = \sum_{k=1}^n (-1)^k \sum_{\alpha_1 > \dots > \alpha_{n-k}} \sum_i \ln\left(1 + e^{c_i + \sum_{\mu=1}^{n-k} W_{i\alpha_\mu}}\right), \tag{35}$$

where $\alpha_\mu \in \{j_1, \dots, j_n\}$.

## C   Analytical calculation of the error of Cossu's pairwise coupling formula

If we rearrange the Eq. (13) with our old parameters $b_j, c_i$ and $W_{ij}$, we can split our pairwise coupling expressions into two different terms with a little algebra as:

$$
\begin{aligned}
J_{j_1 j_2}^{(2)} = \; & \frac{1}{4} \sum_i \ln \frac{\left(1 + e^{c_i + W_{ij_1} + W_{ij_2}}\right)\left(1 + e^{c_i}\right)}{\left(1 + e^{c_i + W_{ij_1}}\right)\left(1 + e^{c_i + W_{ij_2}}\right)} \\[4pt]
& + \frac{1}{4} \sum_i \mathbb{E}_{Y_i^{(j_1 j_2)}} \ln \left[ \frac{\left[1 + \tanh\frac{1}{2}(W_{ij_1} + W_{ij_2} + c_i)\tanh\frac{1}{4}\left(Y_i^{(j_1 j_2)} + \sum_{j \neq j_1, j_2} W_{ij}\right)\right]}{\left[1 + \tanh\frac{1}{2}(W_{ij_1} + c_i)\tanh\frac{1}{4}\left(Y_i^{(j_1 j_2)} + \sum_{j \neq j_1, j_2} W_{ij}\right)\right]} \right. \\[4pt]
& \qquad\qquad \times \frac{\left[1 + \tanh\left(\frac{c_i}{2}\right)\tanh\frac{1}{4}\left(Y_i^{(j_1 j_2)} + \sum_{j \neq j_1, j_2} W_{ij}\right)\right]}{\left[1 + \tanh\frac{1}{2}(W_{ij_2} + c_i)\tanh\frac{1}{4}\left(Y_i^{(j_1 j_2)} + \sum_{j \neq j_1, j_2} W_{ij}\right)\right]} \Bigg] ,
\end{aligned}
\tag{36}
$$

where we have introduced a new variable $Y_i^{(j_1 j_2)}$, defined by

$$Y_i^{(j_1 \dots j_n)} = \sum_{j \neq j_1 \dots j_n} W_{ij} s_j. \tag{37}$$

Now note that the first term of the r.h.s. of Eq. (36) is equivalent to the formula used by Cossu *et al.* [15] for the extraction of 2-body couplings from an RBM trained with Ising model samples. In other words, the expression of Cossu *et al.* ignores the "stochastic" part of the expression.

One can also see that in the case where we consider the particular exact sparse construction discussed in section D for an Ising model with pairwise interaction, in which each hidden node is coupled only to the two spins that interact with each other, the second "stochastic" term in Eq. (36) vanishes exactly, and one recovers the Cossu *et al.* formula.

## D   On the reverse mapping: How can one construct an RBM equivalent to a given generalized Ising Model

Note that the discussion in Section 2.2 primarily concerns situations where we do not know the actual data model and want to use the RBM to infer an effective model from the data. On the other hand, if we already know the physical model that generated the data, it is always possible to find an appropriate set of RBM parameters that reproduce *exactly* your original model. In contrast to the previous case, the mapping in this direction is not unique, many different RBMs

can correspond to exactly the same GIM. Nevertheless, some of these RBMs are much easier to construct by hand than the others. An example of a specific procedure to exactly "train" a very sparse RBM to reproduce up to three body interaction models can be found in Ref. [27].

Let us briefly illustrate in this section how the RBM encodes correlations using hidden variables. In this context, the full interaction model is recovered when we sum out the hidden variables to calculate the marginal probability of the original spin variables. Then, the goal now is to rewrite a GIM Hamiltonian in the form of the marginalized RBM Hamiltonian of Eq. (6). For example, it is easy to verify that the pairwise interaction between $\sigma_1$ and $\sigma_2$ (via $J > 0$) can be rewritten as the marginal energy of these two variables, each interacting lonely with a binary hidden node $\tau(=\pm 1)$ with strength $w$ (satisfying $\cosh 2w = e^{2J}$):

$$\mathcal{H}(\sigma_1, \sigma_2) = -\log \sum_{\tau=\pm 1} e^{-\tilde{\mathcal{H}}(\sigma_1, \sigma_2, \tau)} = -\log\left[2\cosh w(\sigma_1 + \sigma_2)\right] = -J\sigma_1\sigma_2 - J. \quad (38)$$

The right-hand side of the equation results from the even symmetry of the hyperbolic cosine and from the fact that all powers of $\sigma = \pm 1$ are either $\sigma$ or 1. Note that this encoding of $J$ is not unique. Any pair of $w_1$ and $w_2$ that satisfies the conditions $\frac{\cosh(w_1+w_2)}{\cosh(w_1-w_2)} = e^{2J}$ gives the same effective pairwise interaction. This formula can also be used to encode the antiferromagnetic case $J < 0$. The number of possible encodings of $J$ is even larger if we also consider nonzero visible $\eta_1$, $\eta_2$ and hidden $\zeta$ bias in the model.

This construction goes beyond pairwise interactions. A single hidden node interacting with $k$ visible spins introduce $k$-th order interaction and all lowers orders involving all possible combinations of that same set of $k$ spins, i.e.

$$-\log\left[2\cosh\left(\sum_j^k w_j\sigma_j + \zeta\right)\right] = -J^{(k)}_{j_1\ldots j_k}\prod_j^k \sigma_j - \sum_{j=1}^k J^{(k-1)}_{\prod_{l\neq j}}\prod_{l\neq j}^k \sigma_l - \cdots - \sum_{j=1}^k H_j\sigma_j + C, \quad (39)$$

with $C$ a constant. When $\zeta = 0$, only the *even* interaction terms appear in the expansion. Now note that there are $2^k$ ways to arrange $k$ binary spins, which means that (39) leads to a set of $2^k$ linear equations ($2^{k-1}$ if $\zeta = 0$) and $2^k$ possible couplings $J$ or $H$ ($2^{k-1}$ if $\zeta = 0$)[2]. This means that given a set of $\boldsymbol{w}$ and $\zeta$, one can always recover the equivalent interacting spin model, as we derived in the previous section following a simpler strategy than solving the system of linear equations. However, the reverse operation, i.e. choosing a good set of $\boldsymbol{w}$ and $\zeta$ that leads to a desired interaction model, is not possible with only $k + 1$ free parameters. With a little algebra, you can see that with a single hidden node, you can always specify the value of the highest order coupling $J^{(k)}_{j_1\ldots j_k}$ and one or a few other couplings in the expansion, but not the entire set of coupling terms. This means that one would have to add additional hidden nodes to gradually cancel (or correct) the lower order interaction terms. A conservative estimate of the minimum number of hidden nodes required to encode a model with $k$ as the highest interaction order is approximately $N_{\mathrm{h}}^{(k)} = \mathrm{nint}\left[2^k/(k+1)\right]$, where nint is the closest higher integer, or $N_{\mathrm{h}}^{(k)} = \mathrm{nint}\left[2^{k-1}/(k)\right]$ if the interaction model contains only even-order interactions and the hidden biases are set to zero. This is of course a conservative value, since lower-order couplings involving a given set of spins can also occur in the expansion of another hidden node and one only needs to correct each coupling once. Nevertheless, this rough calculation allows us to estimate the order of the number of latent variables required to perfectly encode a given model. It is also clear from the preceding discussion that the number of ways to combine the contributions of different hidden nodes to reproduce a given GIM is very large, especially when the number of hidden nodes used is also large. According to the previous discussion, to perfectly reproduce a model containing up to $N_k$ interactions of order $k$ (where $k$ is the highest order), we need about at least $N_k \times N_{\mathrm{h}}^{(k)}$ hidden nodes.

This rough estimate also gives us an idea of the order of hidden nodes that we should include in an RBM if we want to train it to reproduce a particular model. This brings us back to the question of the number of parameters of such an RBM compared to directly training a GIM with energy function (1). To perfectly fit a model with $N_k$ $k$-body couplings, one would need $(N+1)\times N_k\times N_{\mathrm{h}}^{(k)}$

---

[2]The number of possible couplings at $k - n$ is $\binom{k}{n}$, which means that the total number of couplings that appear in (39) is $2^k = \sum_n^k \binom{k}{n}$.

parameters. If the interactions are sparse and $N_k$ is $O(N)$, the number of parameters scales with $O(N^2)$, which is much lower than the $O(N^k)$ parameters in Eq. (1) for $k > 2$.

It should also be emphasized that low-order couplings can be encoded in many different ways, i.e. a pairwise coupling can be described with a latent variable coupled to the two spins involved (as described above), or it can involve the interaction of several hidden nodes and many spins. In this sense, there are many different ways to encode the same GIM potential (especially when more and more latent variables are added), and the encoding of low-order couplings does not necessarily imply sparse interaction networks between the spins and the latent variables. In fact, the RBMs constructed as described above or in Ref. [27] or the Appendix C, where hidden nodes connect only the visible units that interact directly at each order in the GIM, are much sparser than typical trained RBMs. We show the typical matrices obtained when we train the RBMs with data generated with pairwise models in Fig. 17 in Appendix E, where it is clear that each hidden node is typically connected to many visible units. In Appendix C, we also show that such sparse coupling matrices satisfy our Eq. (13), but the terms involving the average of $X_i^{(j_1 \dots j_2)}$ cancel out.

# E    Validity of the Gaussian Approximation

The approximation scheme proposed in section 2.2 for calculating the expressions (12), (13) and (14) is based on the assumption that the central limit theorem holds for the random variable

$$X_i^{(j_1 \dots j_n)} = \sum_{\mu=n+1}^{N_{\mathrm{v}}} w_{ij_\mu} \sigma'_{j_\mu} = \frac{1}{4} \sum_{\mu=n+1}^{N_{\mathrm{v}}} W_{ij_\mu} \sigma'_{j_\mu}.$$

Here, each $\sigma'_j$ is a stochastic variable that is uniformly distributed over the binary support $\{-1, 1\}$. The conditions under which the central limit theorem holds for this type of variables are discussed, for example, in Appendix B of Ref. [46], which essentially relies on the Lindeberg's theorem. It is not possible to prove this theorem for any RBM, as the structure of the matrix $\boldsymbol{w}$ depends on the particular training method and the data set under investigation.

However, for the RBMs obtained in this work, we observed that there is no particular risk in using this distribution, since we always obtain weights $w_{ij}$ that are essentially Gaussian distributed, as shown in Fig. 17, which means that for small $n$ and large $N_{\mathrm{v}}$ the application of the central limit theorem is safe.

However, in a general case, the entries of $\boldsymbol{W}$ could be very sparse (e.g. if we have constructed the RBM ad hoc using the exact construction of section D) or follow a distribution with a strong tail, in which case a reference to the central limit theorem is certainly very risky. The first case is trivial to treat, since if $W_{ij}$ is only nonzero for a few values of $j$. This means that the contribution of a hidden node $h_i$ to each coupling constant $J_{j_1 \dots j_n}^{(n,i)}$ can be computed in short time. The relevant values of $W_{ij}$ in Eq. (9) are very small. In order to deal with the latter case, an approximation using Large Deviation Theory (see [47] for an understandable introduction) is outlined below.

## E.1    Estimating effective couplings using Large Deviation Theory

First, we introduce a sample mean random variable given by

$$S_i^{(j_1 \dots j_n)} \equiv \frac{1}{N_{\mathrm{v}} - n} \sum_{\mu=n+1}^{N_{\mathrm{v}}} w_{ij} \sigma'_j = \frac{1}{N_{\mathrm{v}} - n} X_i^{(j_1 \dots j_n)}. \tag{40}$$

Let $P_{S_i^{(j_1 \dots j_n)}}(s)$ be the corresponding probability distribution of $S_i^{(j_1 \dots j_n)}$ taking the value $s$. For $N_{\mathrm{v}} - n \gg 1$, one can obtain that

$$P_{S_i^{(j_1 \dots j_n)}}(s) \approx e^{-(N_{\mathrm{v}}-n) I_i^{(j_1 \dots j_n)}}(s), \tag{41}$$

where $I_i^{(j_1 \dots j_n)}(s)$ is the large deviation rate function, which can be calculated by applying the Gärtner-Ellis theorem. To apply this theorem to our case, we compute an estimate of the function

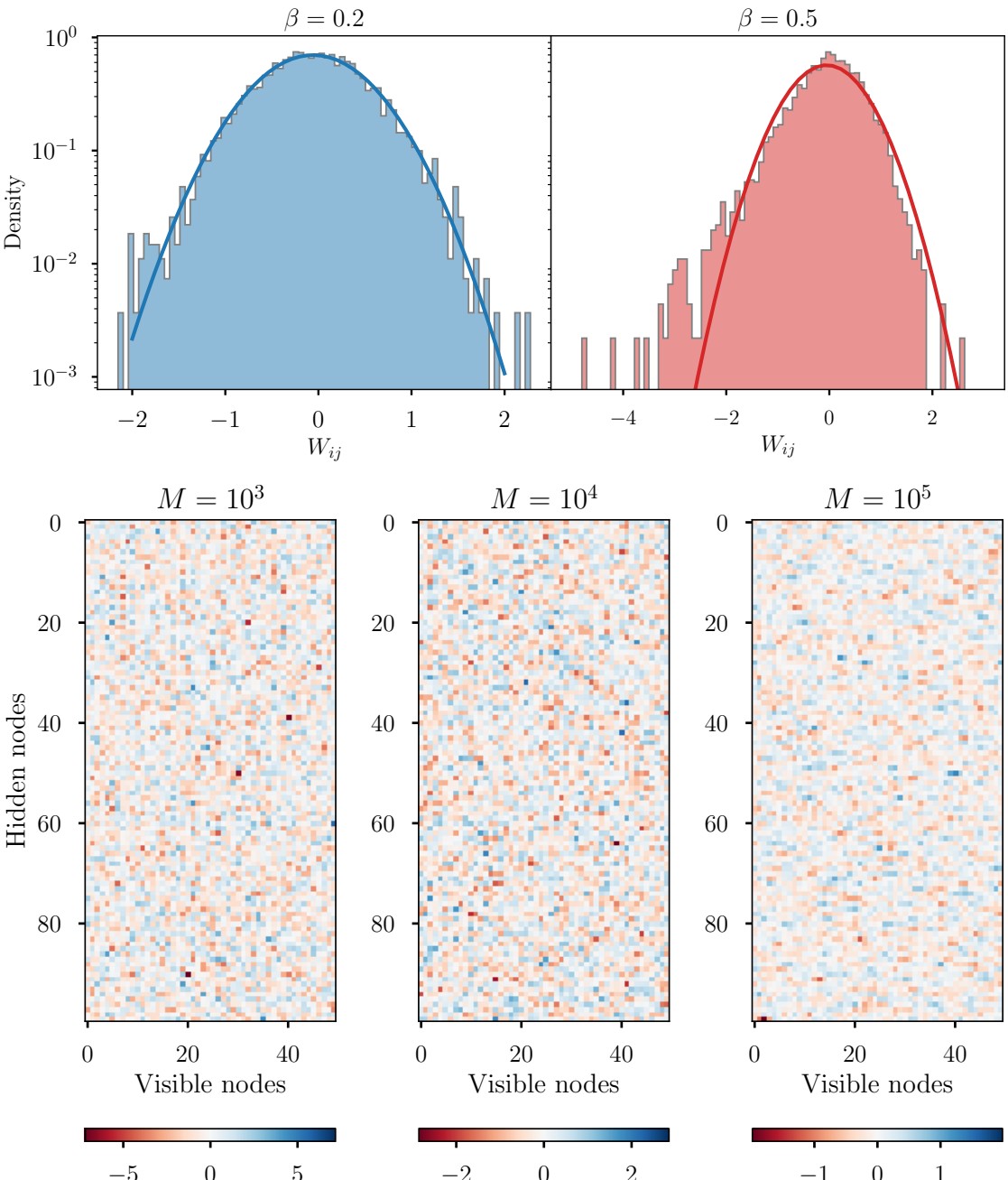

Figure 17: Distribution of the matrix weights of the models trained with samples of the 2D Ising model at $\beta = 0.2,\ 0.5$. Note that in the high-temperature regime, the weight distribution learned by the model is well approximated by a Gaussian function. Otherwise, in the low-temperature regime, one found a heavy-tailed distribution that cannot be well described by a normal distribution. These machines were trained using the PCD-50 scheme, with $N_{\mathrm{h}} = 100, \gamma = 0.01$. The size of the training data in both cases was $M = 10^4$.

$\lambda_i^{(j_1\dots j_n)}(k)$, which is defined as follows

$$
\begin{aligned}
\lambda_i^{(j_1\dots j_n)}(k) &\approx \frac{1}{N_{\mathrm{v}}-n}\ln \mathbb{E}_{S_i^{(j_1\dots j_n)}}\left[e^{(N_{\mathrm{v}}-n)kS_i^{(j_1\dots j_n)}}\right] \\
&= \frac{1}{N_{\mathrm{v}}-n}\ln\left[\left(2^{N_{\mathrm{v}}-m}\right)^{-1}\sum_{\sigma'_{j_{n+1}}=\pm 1}\cdots\sum_{\sigma'_{j_{N_{\mathrm{v}}}}=\pm 1}\exp\left(k\sum_{\mu=n+1}^{N_{\mathrm{v}}}w_{ij_\mu}\sigma'_{j_\mu}\right)\right] \\
&= \frac{1}{N_{\mathrm{v}}-n}\sum_{\mu=n+1}^{N_{\mathrm{v}}}\ln\cosh\left(kw_{ij_\mu}\right).
\end{aligned}
\tag{42}
$$

By applying the Legendre-Fenchel transform to $\lambda_i^{(j_1\dots j_n)}(k)$, we obtain

$$
\begin{aligned}
I(s) &\approx \sup_{k\in\mathbb{R}}\left\{ks - \lambda_i^{(j_1\dots j_n)}(k)\right\} \\
&= \sup_{k\in\mathbb{R}}\left\{ks - \frac{1}{N_{\mathrm{v}}-n}\sum_{\mu=n+1}^{N_{\mathrm{v}}}\ln\cosh\left(kw_{ij_\mu}\right)\right\}.
\end{aligned}
\tag{43}
$$

One could now use this probability $P_{S_i^{(j_1\dots j_n)}}(s)$ to compute the expected values in (12)-(14), instead of taking it as Gaussian distributed. The formulas obtained in this way should in principle be more precise, but we found it difficult to apply them in practice for two reasons. First, reconstructing the couplings in this way takes too long to obtain them systematically along the entire training trajectory, as is done in this work. And second, one sometimes encounters convergence problems when computing the supremum in Eq. (43) numerically. This occurs typically when $I(s)$ is evaluated in a domain that corresponds to regions of low probability mass in the sample space of $S_i^{(j_1\dots j_n)}$, where the function increases monotonically. Therefore, to implement this approximation one should be able to set dynamically the domain of $I(s)$.

# F  Ising Model Simulations

Sampling of the Ising model was performed using various Monte Carlo (MC) algorithms. The Metropolis-Hastings algorithm [48,49] was used for the 1D samples, the Wolff cluster algorithm [50] for the 2D samples, and we used the heat bath to simulate the systems with 3-body interactions. We ensure that the training samples correspond to independent equilibrium configurations of the desired GIM by performing standard time autocorrelation tests (see ref. [51]). The thermalization time before saving the first configuration was set to 1000 or 5000 effective MC steps (EMCS) (we checked that these times are longer than 20 times the exponential autocorrelation time at each temperature), and once the first configuration was saved, we started saving every 100 or 500 EMCS. To verify that the collected samples were independent, we estimated the integrated autocorrelation times $\tau_{\mathrm{int}}$ for each simulation and verified that the number of MC sweeps between two consecutive measurements was much larger than $\tau_{\mathrm{int}}$. Since exact solutions for 1D and 2D ising models are known, we also verified that our simulations were correctly thermalized by ensuring that the expected mean values for a number of observables matched the theoretical expectations (within error). The errors were estimated using the jackknife method. To perform this analysis, we ran a long simulation, $10^6$ or $10^7$ EMCS, through and measured various observables at each EMCS $t$. The observables we analyzed were

- the magnetization $m = \frac{1}{N}\sum_i\langle s_i\rangle$,
- the absolute magnetization $m_{\mathrm{abs}} = \frac{1}{N}\sum_i|\langle s_i\rangle|$,
- the energy per spin $e = \frac{1}{N}\langle\mathcal{H}(\boldsymbol{s})\rangle$,
- and we also checked the Schwinger-Dyson equations for the Ising model [52].

For more details on the thermalization tests see [29].

# G Computational costs

In this section, we discuss the computational cost of training an RBM as well as extracting the effective model parameters and approximating the log-likelihood $\mathcal{L}$ using AIS.

First, it should be noted that the computation time required for RBM training on a GPU scales $\sim (N_\mathrm{v} + N_\mathrm{h})$, since the update of the visible and hidden units can be massively parallelized (all units of a given layer are updated at once). In contrast, the training time of a BM scales $\sim N_\mathrm{v}^2$ because each variable update cannot be parallelized as each variable interacts with all others. This fact makes the training of BMs with maximum likelihood particularly costly compared to RBMs, especially for large system sizes.

In Table 1 we show the physical time required to train an update of the parameters (with $K = 1$) in the RBM (with a NVIDIA Geforce RTX 3090) and in the BM in a processor AMD Ryzen 7 5800× 32GB RAM 2TB, for data with $N_\mathrm{v} = 50$. These times should then be multiplied by $K$ if longer MCMC times were used to estimate the gradient. Our RBMs were trained normally for $10^5$ parameters updates and used $K = 50$ MCMC steps. This means that the total training time for each run reported in this paper was approximately 12 minutes long.

We also show in the table the time needed to compute $\mathcal{L}$ and each 2 and 3 body couplings on a given machine using the Python code in GitHub repository.

Table 1: Empirical Computational times of the various calculations. For RBM/BM parameter updates, the number of parallel chains and MCMC steps was set to $10^3$ and 1 respectively. Similarly, the number of visible variables was $N_\mathrm{v} = 50$. In the RBM case, the number of hidden nodes was $N_\mathrm{h} = 100$. In the approximation of $\mathcal{L}$ with AIS, the number of MC chains and intermediate bridge distributions was $10^4$ and $10^3$ respectively.

| Computation | Empirical time (s) |
|---|---|
| RBM parameter update | $1.4 \times 10^{-4}$ |
| BM parameter update | $1.39 \times 10^{-3}$ |
| $\mathcal{L}$ estimation (AIS) | 2.29 |
| $J^{(2)}$ computation | $1.48 \times 10^{-1}$ |
| $J^{(3)}$ computation | $2.03 \times 10^{-1}$ |

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
