# Peer review of "Inferring effective couplings with Restricted Boltzmann Machines"

_SciPost Physics_

## Round 2 · Referee Report · Anonymous (Referee 1) · 2023-10-29

Strengths

The manuscript “Inferring effective couplings with Restricted Boltzmann Machines” by Decelle et al. describes a quite elegant way to extract effective couplings (local fields, pairwise and higher order couplings) in an Ising spin model from RBM. The authors show that their method, even if it relies on some Gaussian approximation being probably accurate for RBM with quite dense and homogeneous weights, is accurately applicable also to sparsely connected spin systems like 1D and 2D Ising models.

Weaknesses

While the paper is generally interesting, there are some issues, which should be clarified by the authors.

Major comments

  1. The manuscript is an unusual hybrid between a review article and an original article. While the original work is quite problem specific and seems mainly to be of interest for a specialized community, the manuscript starts with a long introduction and review of RBM basics, which are generally known to this community. The first own derivations and results start on page 9. Since I perceive this work mainly as a original contribution, I would recommend the authors to shorten the introduction and to move, if needed, the review about the learning algorithm to an appendix.
  2. The authors argue that RBM have an advantage over pairwise Ising EBM, since they are able to capture higher-order couplings, while Ising models are not. While this is a priori true, I would like to add two remarks:
- In many applications of pairwise models (e.g. in neurosciences and on proteins), the authors check that higher order correlations are reproduced by the model even if not fitted, arguing that in this case a pairwise EBM is sufficient to capture the statistics of the training set.
- RBM contain higher order couplings, but this comes potentially at a high cost in terms of the number of hidden variables. The explicit construction mentioned in the paper already needs O(N^2) hidden nodes for a pairwise model, and since each hidden variable may be connected to all visible variables, we would end up with O(N^3) parameters. This becomes easily unrealistic, in particular if going to higher orders. RBM with fixed number of hidden nodes as studied in the manuscript, have also limted expressive power.
  3. The proposed method is applicable to RBM where both visible and hidden variables are from {0,1}, generalizations to multi-state or continuous variables are not evident. This should be said early.
  4. The reason why the authors transform to Ising spins is not well explained. I guess that the reason is that a pure p-spin model does not generate correlations of orders smaller than p, and thus no effective lower-order couplings. Is this the reason?
  5. I might expect that the Gaussian approximation for the X-variables does not only break down for heavy-tailed w-distributions, but also for sparse weight vectors. Any comments on this? I might imagine that in the 1D and 2D cases, where the underlying models are sparse, also the weight vectors might be sparse. Is this the case? Are the X actually approximately Gaussian distributed?
  6. The coupling error $\Delta_J$ is a very rough and global measure, and hardly interpretable since it does not distinguish between a biased inference of the existing couplings from noise dominated by the large number of zero couplings (I do not even really know if the reported numerical values of the error are large or small in absolute terms). The authors actually do more by plotting coupling histograms, but this could be made a more quantitative information using, e.g., ROC curves. Since apparantly very different models can well fit the same data, estimating some diversity measure of the resulting probability distributions over data space (e.g. D_KL) might also be helpful. It might be particularly helpful when comparing RBM with models trained with BM learning or BP.
  7. In my opinion, something is missing in the 1D and 2D analyses. The authors show that, provided sufficient training data, the fields and two-point couplings are well reproduced by the RBM. But what about higher-order couplings, which may be present in the RBM? Would all J(3) etc. be close to zero? Is the RBM actually able to discover that higher order couplings do not exist? Or does it need larger amounts of training data than needed for getting accurate values of the low-order couplings?
  8. The authors find that, in the case of the 1D model of N=50, about 50 hidden nodes are needed. Do these actually concentrate on individual couplings like in the explicit construction of Sec. 2?
  9. All studied models have simple geometries, being defined on regular (hyper)lattices and having few possible coupling values. Do the findings translate to situations with heterogenous degrees / coordination numbers and real-valued coupling distributions?
  10. The section on non-equilibrium learning is hard to understand, since it uses terms and abbreviations introduced in other articles. While the work on non-equilibrium learning is an interesting research line of the authors, I think it does not contriubute much to the current manuscript.

Minor comments

i) In the beginning section 2.2, the authors say that the RBM distribution Eq. (3) resemble the data distribution. The formulation is a bit sloppy, since the data give only the visible, non the hidden variable. ii) The transformation in Eqs. (13-17) is nice, but why does a simple Taylor expansion of the terms in the effective Hamiltonian (14) does not do the same job? iii) last line of p. 11: should be third-order coupling, not correlation. iv) The 1D model with periodic boundary conditions is not defined on a tree, so BP is not exact, but a probably very good approximation at least for sufficiently large 1D models. As a consequence, BP inference will not become exact for $M\to\infty$ without $N\to\infty$. v) How are the likelihoods calculated? This seems non-trivial due to the presence of the partition function.

Report

The article reports some technically interesting results, but should be reworked in a more concise way. Additional test with better metrics might strengthen the conclusions. In general, also the limits – not only to few data, but also complex distributions – might be better discussed.

---

## Round 2 · Referee Report · Anonymous (Referee 2) · 2023-11-8

Strengths

1 - Novel result further clarifying how to connect couplings with interaction mediated through hidden units.
2 - Preliminary results on simple models demonstrate that the implementation of the method is effective when compared to competitors.

Weaknesses

1 - The paper is intelligible but overall quite lengthy. Working on refocusing the writing would improve the readability. 2 - The equivalence formula can only be evaluated for relatively small n. Which is maybe sufficient for any practical case, but the impact of this limitation is not discussed in the paper. 3 - The stopping criteria implies to track the test likelihood through an Annealed Importance Sampling, which procedure is a priori costly. 4 - The computational cost of the method is not discussed when comparison is made with other methods. 5 - A discussion of possible furthers tests and generalisation is also missing. E.g. the reported experiments are only with +1/-1 couplings, although they include disorder.

Questions: 1 - Add a discussion on computational costs. 2 - Could the author clarify whether all the presented results the RBM training was stopped following the maximum test-likelihood criteria?

Minor: - Page 11 at the end of Section 2: “RBM training is non-convex, which means that many possible RBMs can encode the same data set distribution.’ Such an affirmation is possibly questionable: non convexity means possible local minima, it does not mean that different RBMs can be as good directly, although symmetries in the parametrisation can understandably lead to this conclusion.

Report

The present work derives an equivalence between the learned weights of a Restricted Boltzmann Machine and the many-body couplings of a generalised Ising Models. The derived couplings are expressed using a dummy configuration variable and an effective field. The generic solution for the n-body coupling involving originally a sum over 2^N terms (the size of the configuration space) is re-written as a sum over 2^n terms and an expectation over a random variable involving a sum over all the spins in the system. The trick is then to evaluate the latter expectation by invoking a Central Limit Theorem rather than summing over the 2ˆN terms.

Numerical experiments are used to validate the obtained expression on the 1D Ising, 2D Ising and 2D Edwards-Anderson and a 3-body interaction network. In all cases: (1) a RBM is trained on a dataset generated though Monte Carlo Simulation from the underlying model, (2) the authors check by applying their formula that they recover couplings closely related to the ones from the original model. The authors check that Increasing the number of training samples improves the quality of estimates. They also investigate the influence of the number of hidden units: namely the number of hidden units in the RBM necessary should be equal to the number of non-zero coupling in the model.

Additionally, the authors also compare their approach with other approaches for solving the inverse Ising problem. For pairwise interactions only, a comparable efficiency with Belief Propagation and Boltzmann Machine Learning is found. For higher order interactions, the method of Cossu et al is found to be less precise.

And finally, the authors stress that with out-of equilibrium RBM training, the method is not as accurate as expected, although a reasonable solution could be found at early training times.

Overall, the proposed formula is interesting and the first tests rather convincing. Stating more clearly the (anticipated) limitations and computational issues would be a valuable addition.

---

## Editorial Decision

resubmitted